# Illuminating protein space with a programmable generative model

John B. Ingraham[1], Max Baranov[1], Zak Costello[1], Karl W. Barber[1], Wujie Wang[1], Ahmed Ismail[1], Vincent Frappier[1], Dana M. Lord[1], Christopher Ng-Thow-Hing[1], Erik R. Van Vlack[1], Shan Tie[1], Vincent Xue[1], Sarah C. Cowles[1], Alan Leung[1], João V. Rodrigues[1], Claudio L. Morales-Perez[1], Alex M. Ayoub[1], Robin Green[1], Katherine Puentes[1], Frank Oplinger[1], Nishant V. Panwar[1], Fritz Obermeyer[1], Adam R. Root[1], Andrew L. Beam[1], Frank J. Poelwijk[1] & Gevorg Grigoryan[1✉]

Three billion years of evolution has produced a tremendous diversity of protein molecules[1], but the full potential of proteins is likely to be much greater. Accessing this potential has been challenging for both computation and experiments because the space of possible protein molecules is much larger than the space of those likely to have functions. Here we introduce Chroma, a generative model for proteins and protein complexes that can directly sample novel protein structures and sequences, and that can be conditioned to steer the generative process towards desired properties and functions. To enable this, we introduce a diffusion process that respects the conformational statistics of polymer ensembles, an efficient neural architecture for molecular systems that enables long-range reasoning with sub-quadratic scaling, layers for efficiently synthesizing three-dimensional structures of proteins from predicted inter-residue geometries and a general low-temperature sampling algorithm for diffusion models. Chroma achieves protein design as Bayesian inference under external constraints, which can involve symmetries, substructure, shape, semantics and even natural-language prompts. The experimental characterization of 310 proteins shows that sampling from Chroma results in proteins that are highly expressed, fold and have favourable biophysical properties. The crystal structures of two designed proteins exhibit atomistic agreement with Chroma samples (a backbone root-mean-square deviation of around 1.0 Å). With this unified approach to protein design, we hope to accelerate the programming of protein matter to benefit human health, materials science and synthetic biology.

Protein molecules perform most of the biological functions necessary for life, but creating them is a complicated task that has taken billions of years of evolution. The field of computational protein design aims to shorten this process by automating the design of functional proteins in a programmable manner. Although there has been considerable progress towards this goal over the past three decades[2,3], including the design of previously unknown topologies, assemblies, binders, catalysts and materials[4–7], most de novo designs have yet to approach the complexity and variety of macromolecules that are found in nature. Reasons for this include the fact that modelling the relationship between sequence, structure and function is difficult, and most methods of computational design rely on iterative search and sampling processes that, just like evolution, must navigate a rugged fitness landscape incrementally[8]. Although many computational techniques have been developed to accelerate this search[3] and to improve the prediction of natural protein structures[9], the space of possible proteins remains combinatorially large and is only partly accessible to conventional computational methods. Determining how to efficiently explore the space of designable protein structures remains an open challenge.

An alternative and potentially appealing approach to protein design is to sample directly from the space of proteins that is compatible with a set of desired functions. Although this approach could address the fundamental limitation of iterative search methods, it would require a way to parameterize the a priori 'plausible' protein space, a way to draw samples from this space, and a way to bias this sampling towards desired properties and functions. Deep generative models have proven successful in solving these kinds of high-dimensional modelling and inference problems in other domains, for example in the text-conditioned generation of photorealistic images[10–12]. For this reason, there has been considerable work to develop generative models of protein space, applied to both protein sequences[13–19] and structures[20–26].

Despite recent advances in generative models for proteins, we argue that there are three properties that have yet to be realized simultaneously in one system. These are: modelling the joint, all-atom likelihood of sequences and three-dimensional structures of full protein complexes; achieving this with computation that scales sub-quadratically with the size of the protein system; and enabling conditional sampling under diverse design constraints without retraining. The first of these,

[1]Generate Biomedicines, Somerville, MA, USA. ✉e-mail: ggrigoryan@generatebiomedicines.com

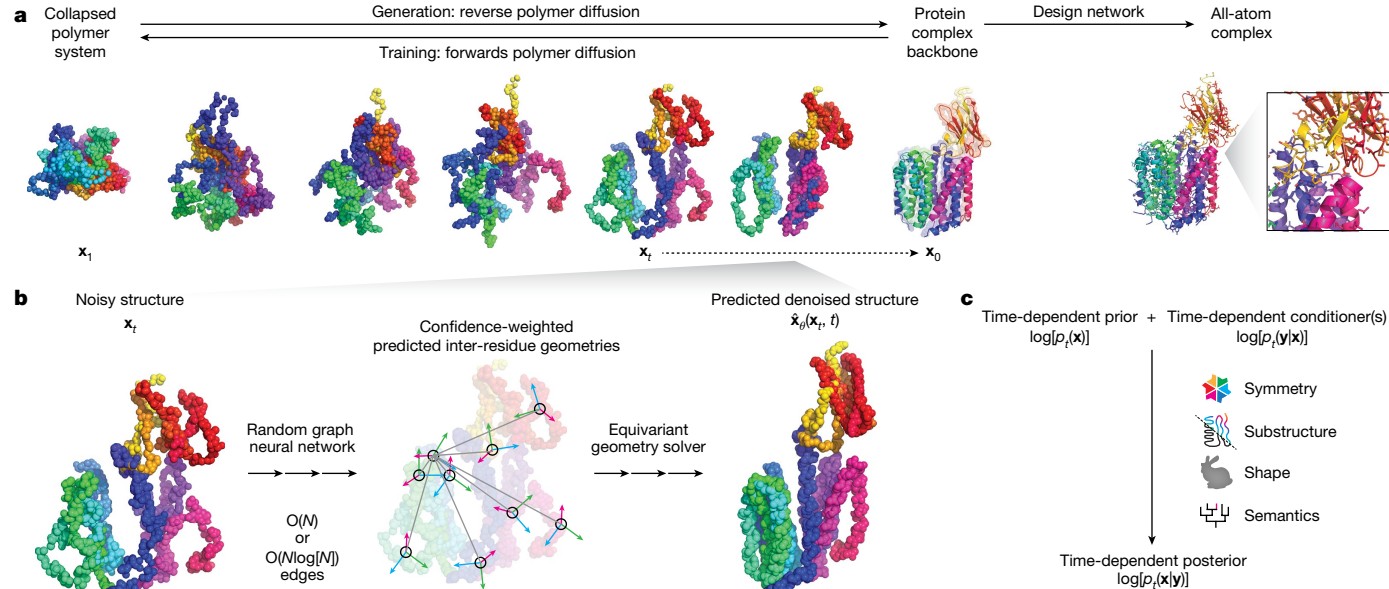

**a** Collapsed polymer system · Generation: reverse polymer diffusion · Training: forwards polymer diffusion · Protein complex backbone · Design network · All-atom complex

$\mathbf{x}_1$ ··· $\mathbf{x}_t$ ·····> $\mathbf{x}_0$

**b** Noisy structure $\mathbf{x}_t$ · Confidence-weighted predicted inter-residue geometries · Predicted denoised structure $\hat{\mathbf{x}}_\theta(\mathbf{x}_t, t)$

Random graph neural network · $O(N)$ or $O(N\log[N])$ edges · Equivariant geometry solver

**c** Time-dependent prior + Time-dependent conditioner(s)
$\log[p_t(\mathbf{x})]$ · $\log[p_t(\mathbf{y}|\mathbf{x})]$

Symmetry
Substructure
Shape
Semantics

Time-dependent posterior
$\log[p_t(\mathbf{x}|\mathbf{y})]$

**Fig. 1 | Chroma is a generative model for proteins and protein complexes that combines structured diffusion for protein backbones with scalable molecular neural networks for backbone synthesis and all-atom design. a**, A correlated diffusion process with chain and radius-of-gyration constraints gradually transforms protein structures into random collapsed polymers (right to left). The reverse process (left to right) can be expressed in terms of a time-dependent optimal denoiser $\hat{\mathbf{x}}_\theta(\mathbf{x}_t, t)$ that maps noisy coordinates $\mathbf{x}_t$ at time $t$ to predicted denoised coordinates $\mathbf{x}_0$. **b**, We parameterize this in terms of a random graph neural network with long-range connectivity inspired by efficient $N$-body algorithms (middle) and a fast method for solving for a global consensus structure given predicted inter-residue geometries (right). Another graph-based design network (**a**, top right) generates protein sequences and side-chain conformations conditionally based on the sampled backbone. **c**, The time-dependent protein prior learnt by the diffusion model can be combined with composable restraints and constraints for the programmable generation of protein systems.

generating full complexes, is important because proteins function by interacting with other molecules, including other proteins. The second, the sub-quadratic scaling of computation, is important because it has been an essential ingredient for managing complexity in other modelling disciplines, such as computer vision, in which convolutional neural networks scale linearly with the number of pixels in an image, and in computational physics, which uses fast $N$-body methods for the efficient simulation of everything from stellar systems to molecular ones[27]. Finally, the requirement to sample from a model without having to retrain it on new target functions is of considerable interest because protein design projects often involve many complex and composite requirements that may vary over time.

Here we introduce Chroma, a generative model for proteins that achieves all three of these requirements by modelling full complexes with quasi-linear computational scaling and by allowing arbitrary conditional sampling at generation time. It builds on the framework of diffusion models[28,29], which model high-dimensional distributions by learning to gradually transform them into simple distributions in a reversible manner, and of graph neural networks[30,31], which can efficiently process geometric information in complex molecular systems. We show that Chroma generates high-quality, diverse and innovative structures that refold both in silico and in crystallographic experiments, and that it enables the programmable generation of proteins conditioned on diverse properties such as symmetry, shape, protein class and even textual input. We anticipate that scalable generative models such as Chroma will enable a widespread and rapid increase in our ability to design and build protein systems that are fit for function.

## A scalable generative model for protein systems

Chroma achieves high-fidelity, efficient generation of proteins by introducing a new diffusion process, neural-network architecture, and sampling algorithm based on principles from contemporary generative

modelling and biophysical knowledge. Diffusion models generate data by learning to reverse a 'noising' process, which for previous image-modelling applications has typically been uncorrelated Gaussian noise. By contrast, our model learns to reverse a correlated noise process to match the distance statistics of natural proteins, which have scaling laws that are well understood from biophysics (Fig. 1a, Supplementary Appendix D). Previous generative models for protein structure have typically leveraged computation that scales quadratically, $O(N^2)$ (refs. 24,25), or cubically, $O(N^3)$ (refs. 9,23), in the number of residues $N$. This has either limited their application to small systems or required large amounts of computation for modestly sized systems. To overcome this problem, Chroma introduces a novel neural-network architecture (Fig. 1b, Supplementary Figs. 4–8, Supplementary Tables 2–3 and Supplementary Appendices E–G) for processing and updating molecular coordinates that uses random long-range graph connections with connectivity statistics inspired by fast $N$-body methods[27] and that scales sub-quadratically ($O(N)$ or $O(N\log[N])$; Supplementary Fig. 4 and Supplementary Appendix E). We found that these modelling components improved performance, as measured by likelihood and in silico refolding across an ablation study of seven different model configurations (Supplementary Fig. 22 and Supplementary Appendix L). Finally, we introduce methods for low-temperature sampling with a modified diffusion process that allows us to trade an increased quality of sampled backbones (increasing likelihood) for reduced conformational diversity (reducing entropy; Supplementary Figs. 1–2, Supplementary Table 4 and Supplementary Appendix C). Given backbones from this diffusion process, the Chroma design network then generates sequence and side-chain conformations that are conditioned on the sampled backbone to yield a joint generative model for the sequences and structure of a protein complex. The design network is based on a similar graph neural-network architecture (Supplementary Figs. 7, 8 and 15), but with conditional sequence and side-chain decoding layers that build on previous studies[15,16] that have seen further refinement and experimental validation[32–34].

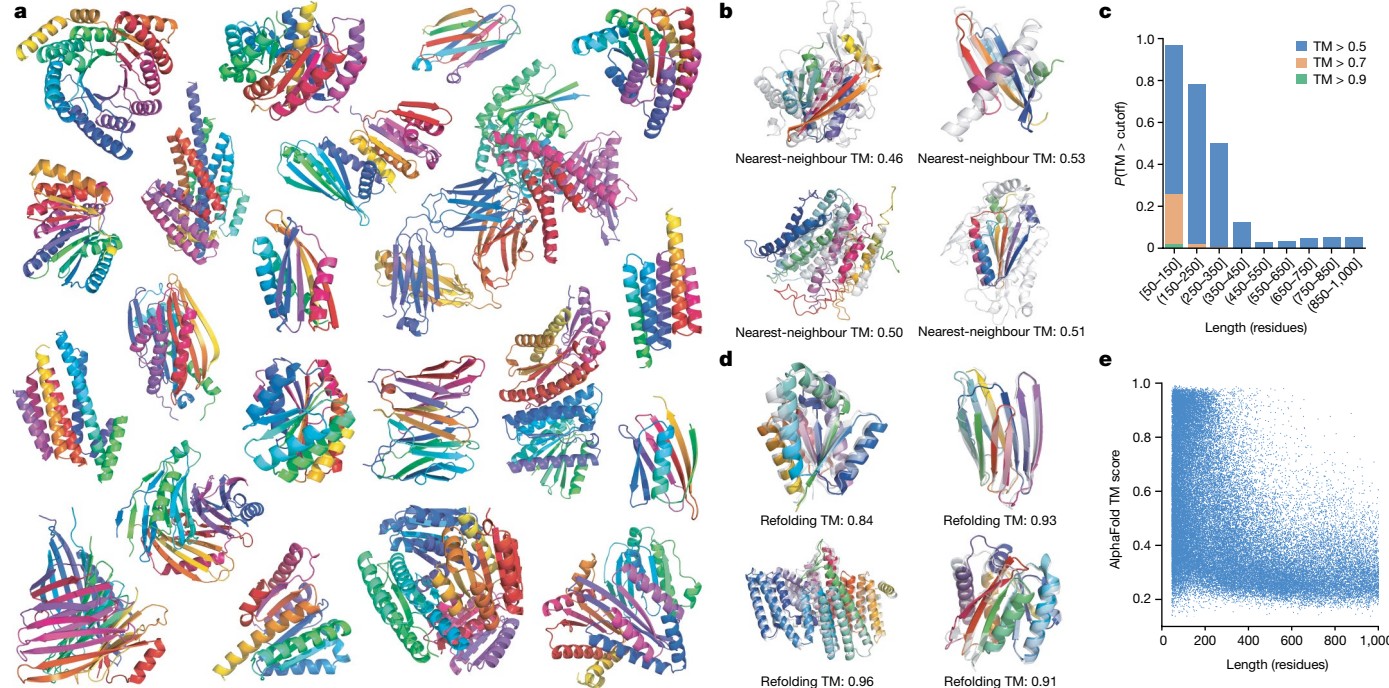

**Fig. 2 | Analysis of unconditional samples reveals diverse geometries that exhibit new higher-order structures and refold in silico. a**, A representative set of Chroma-sampled proteins and protein complexes exhibits complex and diverse topologies with high secondary-structure content, including familiar TIM (triose-phosphate isomerase) barrel-like folds (top left), antibody–antigen-like complexes (centre right) and new arrangements of helical bundles and β-sheets. **b,c**, Despite these qualitative similarities, samples frequently have low nearest-neighbour similarity to structures in the PDB, as measured by nearest-neighbour TM score[41] (**b**; Supplementary Appendix J.4), with structures demonstrating frequent novelty across length ranges (**c**). **d,e**, When we attempted to refold samples in silico using only a single sequence sample per structure, we observed widespread refolding with a high degree of superposition (**d**), including occasionally in the very high length range of more than 800 residues (**e**).

An important aspect of our diffusion-based framework is that it enables programmability of proteins through conditional sampling under combinations of user-specified constraints. This is made possible by a key property of diffusion models: they learn a process that transforms a simple distribution into a complex data distribution through a sequence of many infinitesimal steps. These 'microscopic' steps, therefore, can be biased or constrained by different user-specified requirements to produce a new conditional diffusion process at design time. We built on this with a diffusion-conditioner framework that allows us to automatically sample from arbitrary mixtures of hard constraints and soft penalties implemented as composable primitives (Fig. 1c and Supplementary Appendix M). We explored several conditioner primitives including geometrical constraints that can outfill proteins from fixed substructures (Supplementary Appendix N), enforce particular distances between atoms (Supplementary Appendix O), graft motifs into larger structures (Supplementary Appendix P), symmetrize complexes under arbitrary symmetry groups (Supplementary Appendix Q) and enforce shape adherence to arbitrary point clouds (Supplementary Appendix R). We also explored the possibilities of semantic prompting by training neural guidance networks that predict multi-scale protein classifications (Supplementary Appendix S) and natural language annotations (Supplementary Appendix T) from protein structures. We can invert these predictive models by sampling proteins that optimize classifier predictions. Any subset of conditioners may then be composed for bespoke, on-demand protein generation subject to problem-specific requirements.

## Analysis of unconditional samples

We sought to characterize the space of possible proteins parameterized by Chroma by generating a large number of unconditional samples of proteins and protein complexes (100,000 single-chain proteins and 20,000 complexes across two versions of the models, v.0 and v.1; Supplementary Appendix G and Supplementary Table 2). As can be seen in Fig. 2a, unconditional samples display many properties shared by natural proteins, such as complex layering of bundled α-helices and β-sheets in cooperative, unknotted folds. In some cases, we observed recognizable protein-complex configurations, including what seems to be an antibody–antigen complex in Fig. 2a (centre-right); note that the closest Protein Data Bank (PDB) structural matches to the two 'antigen' chains of this complex are at template-modelling (TM) scores[41] of 0.46 and 0.43, indicating that this sample is not a result of memorization. We provide grids of random samples in Supplementary Figs. 9 and 10 for single-chain and complex structures, respectively. To quantitatively characterize the agreement of Chroma samples with natural proteins, we computed distributions of several key structural properties, including secondary-structure utilization, contact order[35], length-dependent radius of gyration[36], length-dependent long-range contact frequency and density of inter-residue contacts (Supplementary Table 5 and Supplementary Appendix J). We observe a general agreement of these statistics with corresponding distributions from the PDB (Supplementary Fig. 11), although we do see an overrepresentation of α-helices in the later version of Chroma (v.1) that seems to be a consequence of low-temperature sampling, which accentuates the already increased frequency of helices relative to strands in natural proteins (Supplementary Fig. 11b). Because these protein properties focus on low-order structural statistics, we also sought to characterize the extent to which they reproduce higher-order atomic geometries of natural protein structures. Natural protein structures exhibit considerable degeneracy in their use of local tertiary backbone geometries, such that completely unrelated proteins tend to use very similar tertiary motifs[37,38]. Chroma-generated structures exhibit the same type

of degeneracy, utilizing natural tertiary motifs in a way that closely resembles native proteins, including complex tertiary geometries with four or five disjoint backbone fragments (Supplementary Fig. 11c and Supplementary Appendix J).

Although reproducing native-like properties of backbone geometries is important in design, our top priority is the extent to which the proteins can be realized as sequences that fold and function as intended. The definitive answer to this question involves experimental characterization (see below), but in silico evidence can be gathered more systematically. We sought to evaluate the fidelity of sequence–structure pairs generated by Chroma by measuring their agreement with three state-of-the-art methods for structure prediction[9,39,40]. We sampled one sequence for each backbone with Chroma's design network and assessed whether each structure-prediction method would predict these sequences to fold into the corresponding generated structures (Supplementary Fig. 14 and Supplementary Appendix J). We observed widespread refolding of Chroma samples, whether stratified by protein length (Fig. 2e) or helical content and novelty (Supplementary Fig. 14). It is not surprising that successful refolding is less frequent for longer proteins, but it is remarkable that high TM scores[41] are routinely achieved even for proteins more than 800 residues in length. Interestingly, helix content does not seem to be as strong of a predictor of refolding as the distance to the nearest neighbour in the PDB (Supplementary Fig. 15, middle and bottom rows, respectively). We note that this sequence–structure consistency test is not perfect because it rests on the assumption that structure-prediction models will generalize to new folds and topologies. However, the test does provide partial supporting evidence for the generation of realizable protein models in instances in which the predicted and generated structures have strong agreement.

Quantification of the structural homology between Chroma-generated samples and proteins in the PDB indicates that the model generates previously unseen structures at a frequency that increases sharply with length (Fig. 2c and Supplementary Fig. 12a). However, this analysis suffers from the problem that coverage of longer structures is expected to be lower in any finite database. To get a better understanding of the novelty of Chroma samples at different lengths, we defined a novelty score as the number of CATH[42] domains required to greedily cover 80% of the residues in a protein at a TM score above 0.5, normalized by protein length (Supplementary Appendix J). Note that most valid proteins will be covered by at least some finite number of CATH domains because we retain even very small domains (such as single secondary-structural elements) in the coverage test. As shown in Supplementary Fig. 12c,d, there is a clear gap between native and Chroma-generated proteins by this metric, with most native backbones requiring approximately 2–5 times fewer CATH domains to be covered per length than generated backbones.

We also find that samples from Chroma are diverse and cover natural protein space. In Supplementary Fig. 13, we present samples from Chroma and a set of native structures with global topology descriptors derived from knot theory[43,44] and embed them into two dimensions with UMAP[45]. The resulting embedding seems to be semantically meaningful because subsets of structures belonging to different categories by size and secondary structures cluster in this projection (sub-panels on the left in Supplementary Fig. 13a). False colour of the points in the embedding shows that novelty is spread broadly and is not biased to only certain types of structure space. This is especially clear when looking at a representative selection of samples shown in Supplementary Fig. 13b.

## Programmability

An important aspect of Chroma is its programmability, which means it is straightforward to specify high-level desired protein properties (such as symmetry groups) that are compiled into a set of sampling conditioners that bias the diffusion process towards these properties (Fig. 1c, Supplementary Fig. 23 and Supplementary Appendix M). To

demonstrate the range of protein properties that can be programmed with conditional generation, we explored several composable conditioning primitives (Supplementary Table 6, Supplementary Figs. 23–33 and Supplementary Appendices N–T). Although we believe that each of these represents only a preliminary demonstration of possible conditioning modes, they provide a glimpse of the potential for programmable protein design.

We began by considering analytic conditioners that can control protein backbone geometry. We found that conditioning on the symmetry of protein complexes can readily generate samples under arbitrary symmetry groups (Fig. 3a, Supplementary Figs. 17, 27–29 and Supplementary Appendix Q). Figure 3a illustrates symmetry-conditioned generation across many groups, from simple four-subunit cyclic symmetries up to a capsid-sized icosahedral complex with 60,000 total residues and more than 240,000 atoms. This also demonstrates why favourable computational scaling properties, such as quasilinear computation time (Supplementary Appendix E), are important, as efficient computation enables scaling to larger systems. Symmetric assemblies are common in nature and there have been some successes with de novo symmetric designs[46,47], but it has generally been difficult to simultaneously optimize for both the desired overall symmetry and the molecular interaction details between protomers. Symmetry conditioning within the generation process in Chroma should make it simpler to sample structures that simultaneously meet both requirements.

We next explored substructure conditioning (Fig. 3b, Supplementary Figs. 16, 24–26, Supplementary Appendices N–P), which is a central problem for protein design because it can enable the preservation of one part of the structure of a protein (such as an active site) while modifying another part of the structure (and potentially function). In the top row, we cut the structure of human dihydrofolate reductase (DHFR; PDB code 1DRF) into two halves with a plane, remove one of the halves and regenerate the missing half. The cut plane introduces multiple discontinuities in the chain simultaneously, and the generative process must sample a solution that simultaneously satisfies these boundary conditions while being biophysically plausible. Nevertheless, the samples achieve both goals and, interestingly, do so in a manner very different from each other and from natural DHFR. In the second row of Fig. 3b, we cut out the complementarity-determining regions of a VHH antibody and rebuilt them conditioned on the remaining framework structure. Finally, the bottom three rows of Fig. 3b condition on sub-structure in an unregistered manner, meaning that the exact alignment of the substructure (motif) within the chain is not specified a priori, as it was in the previous examples. We outfilled the protein structure around several structural and functional motifs, including an αββ packing motif, backbone fragments encoding the catalytic triad active site of chymotrypsin and the EF-hand Ca-binding motif. Again, these motifs are accommodated in a realistic manner using diverse and structured solutions.

In Fig. 3c we provide an early demonstration of a more exotic kind of conditioning in which we attempted to solve for backbone configurations subjected to arbitrary volumetric shape specifications. We accomplished this by adding heuristic classifier gradients based on optimal transport distances[48] between atoms in the structures and user-provided point clouds (Supplementary Appendix R). As a stress test of this capability, we conditioned the generation of single protein chains on the shapes of the Latin alphabet and Arabic numerals (Supplementary Fig. 18 and Supplementary Appendix K.3). We see the model routinely implementing several core phenomena of protein backbones, such as high secondary-structure content, close packing with room for designed side chains, and volume-spanning α-helical bundle and β-sheet elements. Although these shapes represent purely a challenging set of test geometries, more generally shape is intimately related to function in biology, for example, with membrane transporters, receptors and structured assemblies that organize molecular events in space. Being able to control shape would be a useful subroutine for generalized programmable protein engineering.

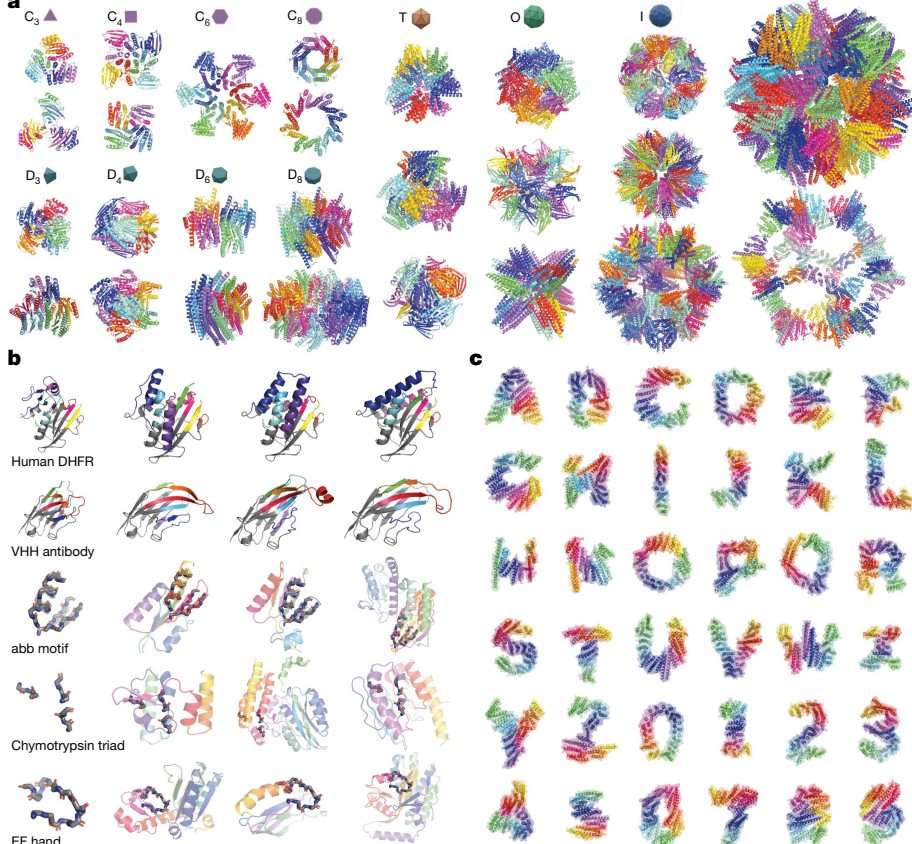

**Fig. 3 | Symmetry, substructure and shape conditioning enable geometric molecular programming. a**, Sampling oligomeric structures with arbitrary chain symmetries is possible by using a conditioner that tessellates an asymmetric subunit in the energy function. Cyclic ($C_n$), dihedral ($D_n$), tetrahedral (T), octahedral (O) and icosahedral (I) symmetry groups can produce a wide variety of possible homomeric complexes. The right-most protein complex contains 60 subunits and 60,000 total residues, which is enabled by leveraging symmetries and using our subquadratically scaling architecture. **b**, Conditioning on partial substructure (monochrome) enables protein infilling or outfilling. The top two rows illustrate regeneration (colour) of half a protein (the enzyme DHFR, first row) or complementarity-determining region loops of a VHH antibody (second row). The next three rows show conditioning on a predefined motif. The order and matching location of motif segments is not prespecified here. **c**, Conditioning on arbitrary volumetric shapes is exemplified by the complex geometries of the Latin alphabet and Arabic numerals. All structures were selected from protocols with high rates of in silico refolding (Supplementary Appendix K).

Finally, we demonstrate in Fig. 4 that it is possible to condition on protein semantics, such as secondary structure, fold class (Fig. 4a, Supplementary Figs. 19, 30 and Supplementary Appendix S) and natural language (Fig. 4b, Supplementary Figs. 20, 31–33, and Supplementary Appendix T). Unlike geometric conditioning, in which the classifier is correct by construction (for example, the presence of a motif with less than a certain root-mean-square deviation is unambiguous), here the classifiers are neural networks trained on structure data, so there can be a discrepancy between the label assigned by the classifier and the ground truth class. Thus, for the fold-conditioned generation (Fig. 4a), we see that conditional samples always improve classifier probabilities over unconditional samples taken from the same random seed, but the classification is not always perfect. For example, for the 'Rossman fold' class, the generated samples reproduce the canonical mixed topology. However, in the 'Ig fold' and 'β-barrel fold' examples, the structures exhibit some of the features characteristic of the classes (for example, β-sheets packed against each other) but do not contain all such features (for example, the Ig topology does not appear canonical and the barrel does not form a closed cycle). In Fig. 4b we demonstrate two examples of semantic conditioning on natural language captions, where we again occasionally observe alignment between samples and intended prompts, especially for highly-represented protein classes. It is exciting to imagine the potential of such a capability, that is being able to request desired protein features and properties directly through natural language prompts. Generative models such as Chroma can reduce the challenge of function-conditioned generation to the problem of building accurate classifiers for functions given structures. Although there is clearly much more work to be done to make this useful in practice, high-throughput experiments and evolutionary data are likely to enable this in the near term.

Supplementary Appendix K demonstrates extensive in silico refolding studies of samples generated with the conditioners described above. As shown in Supplementary Figs. 16–20, all of these conditional-generation processes can produce samples that refold accurately to their generated backbones. The rates at which this happens vary according to the specific condition and protein length (and are subject to the caveats of this test mentioned above), but even in the challenging cases of shape-, complex symmetry-, class- and language-conditioned designs, we observe widespread refolding across specific conditions and structure prediction methods.

## Experimental validation

To experimentally validate Chroma, we built a simple design protocol (based on Chroma v.0) that was intended to generate high-likelihood samples drawn from the model. Specifically, the protocol involved

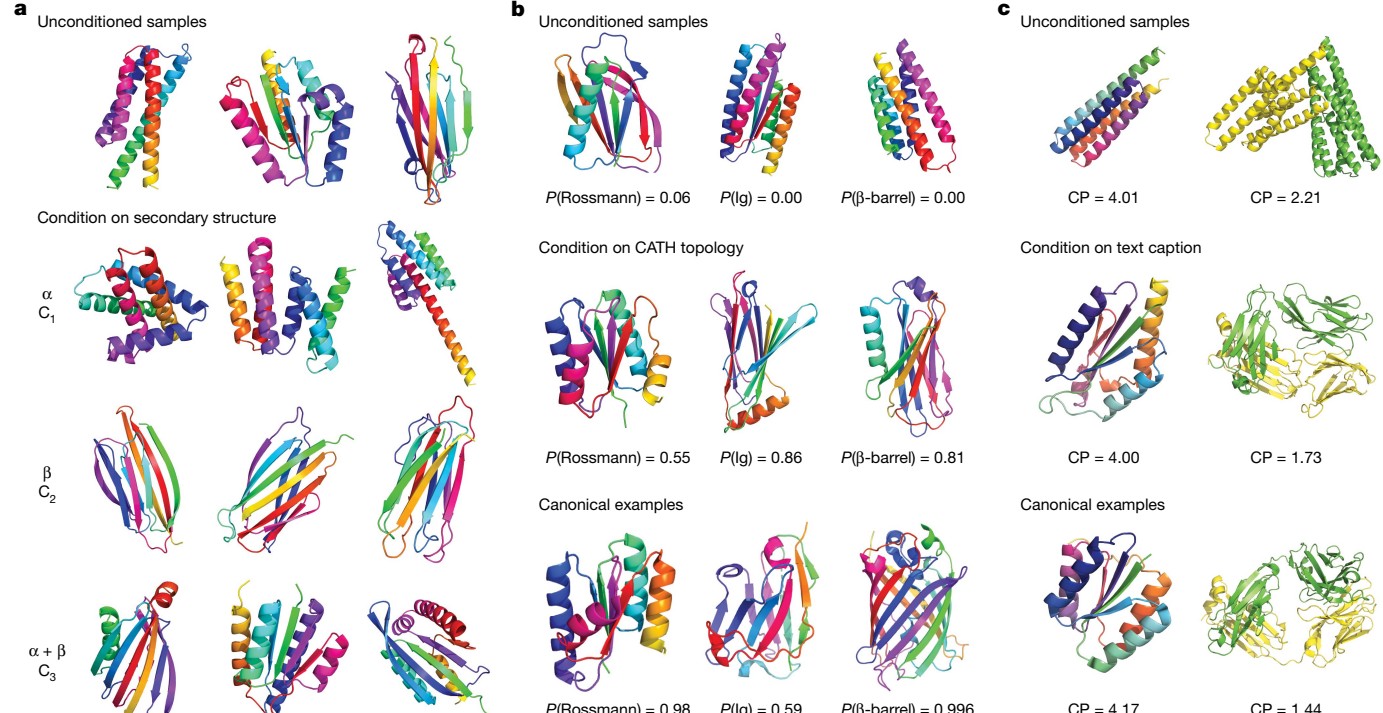

**a** Unconditioned samples

Condition on secondary structure

α
C1

β
C2

α + β
C3

**b** Unconditioned samples

$P$(Rossmann) = 0.06   $P$(Ig) = 0.00   $P$(β-barrel) = 0.00

Condition on CATH topology

$P$(Rossmann) = 0.55   $P$(Ig) = 0.86   $P$(β-barrel) = 0.81

Canonical examples

$P$(Rossmann) = 0.98   $P$(Ig) = 0.59   $P$(β-barrel) = 0.996

**c** Unconditioned samples

CP = 4.01   CP = 2.21

Condition on text caption

CP = 4.00   CP = 1.73

Canonical examples

CP = 4.17   CP = 1.44

**Fig. 4 | Protein structure classifiers and caption models can bias the sampling process towards user-specified properties. a**, Neural networks trained to predict protein properties can bias unconditional samples (top) towards states that optimize predicted properties, such as secondary-structure composition (bottom) indicated by CATH class level codes (C1, Mainly Alpha; C2, Mainly Beta; C3, Alpha Beta). **b**, A neural network trained to predict CATH topology annotations can routinely drive generation towards samples with high predicted probabilities of the intended class label, which sometimes aligns with our intended fold topology for highly abundant labels. Left, highly abundant Rossmann fold (CATH topology 3.40.50, 14.0% of training set); middle,

highly abundant Ig fold (CATH topology 2.60.40, 9.8% of training set); right, a rare specific β-barrel fold (CATH topology 2.40.155, 0.07% of training set). **c**, Fine-tuning a multi-label predictor to bias a pretrained large language model into a structure caption predictor can enable natural language conditioning. We begin to see examples of semantic alignment between prompts and output structures for highly abundant classes of structures, although we do not always see this reflected in the time-zero caption perplexity (CP, lower is better). Left, 'crystal structure of a Rossmann fold'; right, 'crystal structure of a Fab antibody fragment'.

three steps: generate backbones by drawing independent samples from Chroma at low temperature; design sequences for each backbone using the Chroma design network; and automatically select a subset for experimental characterization to match the desired experimental scale, driven primarily by sequence and/or structure likelihood (as shown in Supplementary Table 7 and Supplementary Appendix U.1). Notably, we deliberately did not filter designs for refolding by a structure-prediction method or using any structure–energetic calculations. However, such filtering could potentially be used to improve the success rate of design.

We generated 310 proteins (unconditional or semantically conditioned on CATH class or topology) for attempted expression and structural characterization (Fig. 5a). We first addressed an initial set of 172 unconditional proteins, ranging between 100 and 450 amino acids in length (Supplementary Fig. 36). We used a pooled protein solubility assay that was based on the split-GFP reporter system[49] to prioritize tractable proteins for subsequent characterization (Supplementary Fig. 38a). After FACS and Nanopore sequencing (Supplementary Fig. 38b), enrichment scores were assigned to categorize the soluble expression levels of each protein (Supplementary Fig. 38c). All 172 tested proteins were assigned higher enrichment scores than the negative control (human $β_3$ adrenergic receptor, Supplementary Table 8), indicating that a wealth of Chroma-designed unconditional proteins can be solubly expressed in *Escherichia coli* (Fig. 5b). We confirmed stable fluorescence in sorted cell populations (Supplementary Fig. 38d) and corroborated our split-GFP screen results using western blotting, observing soluble expression of 19 of the 20 top-scoring proteins and 0

of the 20 lowest-scoring proteins (Supplementary Fig. 39). We created an additional set of 96 unconditional Chroma proteins encompassing a wider range of lengths (from 100 to 950 amino acids; Supplementary Fig. 40a), which performed similarly to the first unconditional protein set using the split-GFP reporter assay (Supplementary Fig. 40b,c). In this additional set, soluble expression of nine of the ten top-scoring proteins was confirmed by western blotting (Supplementary Fig. 40d).

Of the proteins identified in the top 10% of the split-GFP solubility screen, we purified seven for interrogation using circular dichroism (CD; Fig. 5e) and differential scanning calorimetry (Supplementary Fig. 41 and Extended Data Table 1). The results indicate that most of the isolated proteins were stably folded with appreciable secondary structure. From these proteins, we were able to obtain X-ray crystal structures (Extended Data Table 2) for UNC_079 (PDB 8TNM; Fig. 5c) and UNC_239 (PDB 8TNO; Fig. 5d). The observed structures matched the anticipated designs to a high degree (root-mean-square deviation = 1.1 Å and 1.0 Å, respectively), indicating that Chroma-generated structures are realizable. Importantly, these structures are unique with respect to the PDB, with the top PDB hit to UNC_079 (PDB entry 4NH2, chain E) having query and target TM scores of 0.7 and 0.3, respectively, and the top hit to UNC_239 (PDB entry 6AFV, chain A) having query and target TM scores of 0.5 and 0.23, respectively (Fig. 5c,d).

The results of the split-GFP assay show that it is more difficult to succeed with longer designs, because there is an inverse correlation between length and split-GFP score (Supplementary Fig. 34). Interestingly, although we might expect the extent of refolding by structure prediction to also correlate with experimental success, we saw

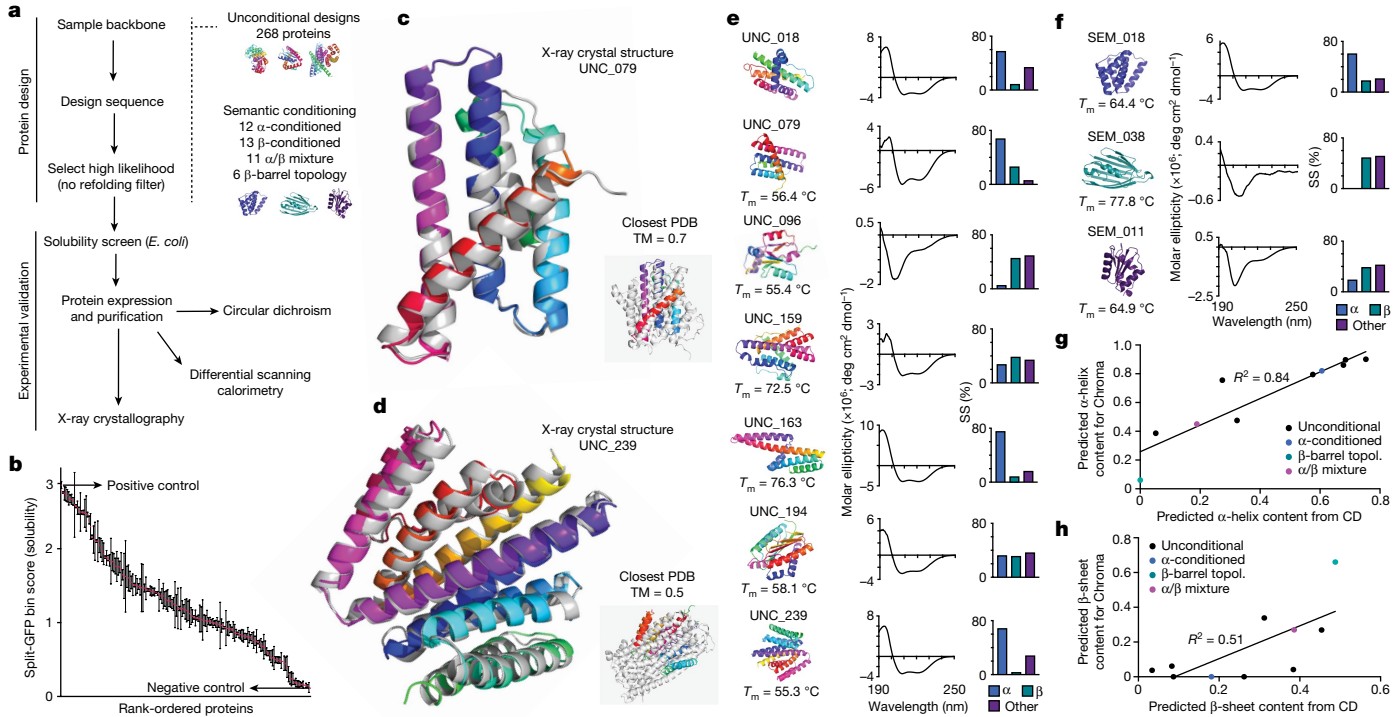

**Fig. 5 | Experimental validation of Chroma-designed proteins. a**, Protocol for protein design and experimental validation. Unconditional designs: 268 proteins. Semantic conditioning: 12 α-conditioned, 13 β-conditioned, 11 α/β mixtures and 6 with β-barrel topology. See text for details. **b**, Rank-ordered unconditional Chroma protein solubility scores by the split-GFP assay for 172 tested proteins. Red dots and error bars denote means and standard deviations, respectively, from three biological replicates. **c,d**, X-ray crystal structures (rainbow) of UNC_079 (**c**, 1.1 Å resolution, PDB 8TNM, root-mean-square deviation (RMSD) = 1.1 Å) and UNC_239 (**d**, 2.4 Å resolution, PDB 8TNO, RMSD = 1.0 Å) overlaid with Chroma-generated models (grey). Insets

compare each crystal structure (rainbow) with its nearest PDB match (4NH2 and 6AFV, respectively; grey). **e**, CD data for seven purified Chroma proteins. The fraction of α-helical and β-strand content was determined using BeStSel[50]. $T_m$ is the melting temperature determined by differential scanning calorimetry and SS designates secondary structure. **f**, CD data for three purified Chroma conditional designs: SEM_018 (α-conditioned), SEM_038 (β-barrel topology) and SEM_011 (α/β mixture). **g,h**, Correlation between predicted secondary-structure content in Chroma designs compared with the prediction from CD, for α-helical (**g**) and β-strand (**h**) content.

no correlation when length is corrected for (Supplementary Fig. 34). Similarly, we saw no correlation between soluble expression and structural novelty. We did find model likelihoods to be weakly predictive of experimental success for the first conditional set, but this did not hold true for the second set, in which lengths were extended up to 950 amino acids (Supplementary Fig. 35).

To test the ability of Chroma to propose well-behaved proteins in a conditioned setting, we next evaluated a set of 42 proteins conditioned by ProClass on CATH class (36 designs split among the classes mainly α, mainly β and mixed α/β) and on CATH topology (six designs conditioned on the β-barrel topology 2.40.155; Supplementary Fig. 37a). In the split-GFP solubility assay, 40 of these proteins (95%) scored above the negative control, indicating a high success rate of soluble protein expression (Supplementary Fig. 37b). We purified one representative protein from each secondary-structure category (two designs conditioned on mainly-α and mixed α/β classes, and one design conditioned on the β-barrel topology). Differential scanning calorimetry data for these proteins were consistent with relatively stable folding, with melting temperatures ranging from 64 °C to 78 °C (Supplementary Fig. 37c). On the basis of secondary-structure predictions from CD spectra[50], we observed higher α-helical content in the mainly-α design, higher β-sheets in the β-barrel design, and mixed secondary structure in the mixed-content protein (Fig. 5f). Indeed, across both conditional and unconditional designs, the inferred secondary-structure content from CD was closely correlated with the secondary-structure content

calculated from Chroma-generated models, for both the fraction of α-helices ($R^2 = 0.84$; Fig. 5g) and β-sheets ($R^2 = 0.51$; Supplementary Fig. 5h), indicating that proteins with various structural compositions can be designed by Chroma.

## Discussion

In this work we present Chroma, a generative model that can generate new and diverse proteins across a broad array of structures and properties. Chroma is programmable in the sense that it can sample proteins with a wide array of user-specified properties, including inter-residue distance and contact, domain, sub-structure and semantic specification from classifiers. Chroma is able to generate proteins that have arbitrary and complex shapes, and it has even begun to demonstrate the ability to accept descriptions of desired properties as free text. Its efficient design, with an innovative diffusion process, quasilinear scaling neural architecture and low-temperature sampling method, means that Chroma can generate extremely large proteins and protein complexes (with more than 3,000 residues) on a commodity graphics processing unit (such as an NVIDIA V100) in a few minutes.

We reasoned that the best way to determine the plausibility of the protein space parameterized by Chroma was to draw independent samples from the model and test them experimentally. Note that this is a departure from the prototypical protein-design protocol, in

which initial proposal designs are down-selected using a custom set of filters intended to avoid known or hypothesized model deficiencies and help focus on designs that are more likely to work experimentally. Although the latter practice, which is broadly adopted in the field, can be effective at increasing design success rates, it does require a custom set of filters for each design project and makes fully automated design difficult to achieve. Furthermore, such an approach would detract from our intention of characterizing the distribution learned by Chroma.

Our experimental validation shows that Chroma has learnt a sufficiently accurate distribution such that sampling from it results in proteins that express, fold, have favourable biophysical properties and conform to intended structures at non-trivial rates. Even under the highly conservative view that only the proteins we purified and characterized individually in solution constitute successful designs (as opposed to others that performed comparably by split-GFP, for example), Chroma would still have a 3% success rate. Moreover, the two designs with experimentally determined crystal structures demonstrate that a non-trivial fraction of this distribution should be expected to be atomistically accurate. Given the breadth and novelty of the structure space learned by Chroma (Fig. 2 and Supplementary Figs. 9, 10 and 13), even these conservative estimates of success rate would translate into immense swaths of unexplored actionable protein space that can now be accessible through commodity computing hardware.

The task of exploring protein structure space in a way that can produce physically reasonable and designable conformations has been a long-standing challenge in protein design. In a few protein systems, it has been possible to parameterize the backbone conformation space mathematically—most notably the α-helical coiled coil[51] and a few other cases that have high symmetry[52]—and in these cases, design efforts have benefited tremendously, creating possibilities that are not available in other systems[52,53]. For all other structure types, however, a great amount of computational time has been spent on the search for reasonable backbones, often leaving the focus on actual functional specifications out of reach. Chroma has the potential to address this problem, enabling a shift from focusing on generating feasible structures towards a focus on the specific task at hand—namely, what the protein is intended to do. By leveraging proteins sampled over more than 3 billion years of evolution, and by finding new ways to assemble stable protein matter, generative models such as Chroma are well poised to drive another expansion of biomolecular diversity with benefits for human health and bioengineering.

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

## Reporting summary

Further information on research design is available in the Nature Portfolio Reporting Summary linked to this article.

## Data availability

All experimental and computational results are available in the Supplementary Information and Extended Data Tables 1 and 2. Experimental structures solved as part of this study were deposited under PDB accession codes 8TNM and 8TNO. Training datasets were constructed based on the PDB (https://www.rcsb.org/), as queried on 20 March 2022, UniProt 2022_01 (https://www.uniprot.org) and PFAM 35 (http://pfam.xfam.org/). PDB IDs comprising Chroma training, test and validation sets are available in the Zenodo dataset at https://doi.org/10.5281/zenodo.8285077.

## Code availability

Chroma code is available at https://github.com/generatebio/chroma under the Apache 2.0 open-source licence.

**Acknowledgements** We thank W. F. DeGrado, R. Kormos and Generate employees A. Ramos, A. Delhagen, A. Jecrois, B. R. P. Saravanan, B. Hannigan, B. Patuto, B. Vogler, D. Moonan, D. Curran, D. Ferguson, E. Brignole, E. Palovcak, J. Lucas, J. McFarland, J. Huaman-Argandona, J. Garlick, K. Tamang, K. Hopson, M. Pattie, M. Jankowiak, M. Saputo, M. Nally, M. Mathur, M. Gibson, N. Shaban, N. Joh, R. Chaudhary, R. Federman, S. Clancy, S. DeCamp, T. Linsky, Y. Liu and Z. Harteveld for assistance with experimental and computational methods development, discussions and input on manuscript drafts; and B. Turner and staff at the MIT Biophysical Instrumentation Facility for providing training and access to the CD spectrometer. The study used the resources of the MIT Structural Biology Core Facility and the MIT Biophysical Instrumentation Facility.

**Author contributions** J.B.I. and G.G. led the research. J.B.I. developed the generative model. J.B.I., M.B., Z.C., W.W., A.I. and F. Obermeyer developed the conditioner and sampling ecosystem. J.B.I., M.B., F. Oplinger, V.F., V.X. and G.G. built and applied the infrastructure for training and inference. J.B.I., K.W.B., A.L.B., F.J.P. and G.G. wrote the manuscript. J.B.I., M.B., Z.C., K.W.B., W.W., A.I., V.F., D.M.L. and G.G. wrote the Supplementary Information. J.B.I., M.B., Z.C., W.W., A.I., V.F., V.X., S.T. and G.G. conducted in silico experiments. F.J.P. conceived and supervised the experimental characterization plan. K.W.B., C.N.-T.-H., E.R.V.V., K.P., N.V.P., A.L. and S.C.C. designed and performed high-throughput experimental studies. K.W.B., J.V.R. and A.M.A. designed and performed biophysical studies. D.M.L., C.N.-T.-H., E.R.V.V. and C.L.M.-P. performed structural characterization. D.M.L., K.W.B. and R.G. designed constructs and performed protein expression. J.B.I., A.R.R., A.L.B., F.J.P. and G.G. supervised the research.

**Competing interests** All authors are employees and shareholders of Generate Biomedicines.

**Additional information**
**Correspondence and requests for materials** should be addressed to Gevorg Grigoryan.

# Article

**Extended Data Table 1 | Differential scanning calorimetry data**

| Protein | DSC | | | | | | | | | | |
|---------|---------|---------------|-----------------------|-----------|-------------|----------|-----------------------|-----------------------|----------|-----------------------|-----------------------|
|         | Quality | Model | Total Area (kJ/mole) | Conc (µM) | Tonset (ºC) | Tm1 (ºC) | ΔH cal 1 (kJ/mol) | ΔH VH 1 (kJ/mol) | Tm2 (ºC) | ΔH cal 2 (kJ/mol) | ΔH VH 2 (kJ/mol) |
| UNC_018 | No Signal | No Fit | -- | -- | -- | -- | -- | -- | -- | -- | -- |
| UNC_079 | Good | Non-Two State | 104 | 21.9 | **39.46** | **56.41** | 105 ± 1.37 | 231 ± 3.72 | | | |
| UNC_096 | Good | Non-Two State | 104 | 24.2 | **37.30** | **55.42** | 103 ± 3.27 | 224 ± 8.76 | -- | -- | -- |
| UNC_159 | Good | Non-Two State | 188 | 18.6 | **48.45** | **72.49** | 189 ± 1.69 | 165 ± 1.7 | **85.90** | 7.99 ± 0.92 | 596 ± 71.3 |
| UNC_163 | Good | Non-Two State | 282 | 20.0 | **62.66** | **76.31** | 289 ± 0.69 | 338 ± 0.99 | | | |
| UNC_194 | Good | Non-Two State | 151 | 13.0 | **41.15** | **58.07** | 157 ± 0.68 | 199 ± 1.06 | | | |
| UNC_239 | Good | Non-Two State | 423 | 15.6 | **30.19** | **55.26** | 431 ± 2.79 | 113 ± 0.74 | **67.63** | 20.3 ± 1.63 | 358 ± 29.7 |
| SEM_011 | Low Signal | Non-Two State | 70.9 | 36.4 | **33.69** | **64.37** | 73.5 ± .61 | 133 ± 1.37 | | | |
| SEM_018 | Good | Non-Two State | 273 | 29.3 | **64.88** | **77.77** | 279 ± 0.70 | 351 ± 1.09 | | | |
| SEM_038 | Good | Non-Two State | 281 | 13.4 | **54.33** | **64.94** | 131 ± 1.39 | 330 ± 4.26 | **73.92** | 159 ± 0.96 | 782 ± 4.85 |

Differential scanning calorimetry data for Chroma proteins evaluated experimentally.

**Extended Data Table 2 | X-ray crystallography data collection and refinement statistics**

| | UNC_079 | UNC_239 |
|---|---|---|
| **Data Collection** | | |
| PDB ID | 8TNM | 8TNO |
| Space group | P $4_3$ $2_1$ 2 | P $2_1$ $2_1$ $2_1$ |
| *Cell dimensions* | | |
| a, b, c (Å) | 59.54, 59.54, 89.18 | 41.81, 80.61, 164.30 |
| α, β, γ (Å) | 90.00, 90.00, 90.00 | 90.00, 90.00, 90.00 |
| Resolution | 35.69 - 1.10 (1.13 - 1.10) | 45.30 - 2.36 (2.42 - 2.36) |
| Total Reflections | 810223 (19457) | 173460 (13300) |
| Unique Reflections | 65204 (4392) | 23239 (1716) |
| I/σI | 15.2 (0.4) | 12.2 (1.8) |
| Completeness (%) | 99.2 (91.7) | 98.2 (99.9) |
| Redundancy | 12.4 (4.4) | 7.5 (7.8) |
| $R_{merge}$ | 0.053 (2.34) | 0.067 (0.931) |
| $R_{meas}$ | 0.057 (2.90) | 0.072 (0.999) |
| $R_{pim}$ | 0.021 (1.68) | 0.026 (0.348) |
| | | |
| **Refinement** | | |
| Resolution (Å) | 35.69 - 1.10 (1.12 - 1.10) | 45.29 - 2.36 (2.44 - 2.36) |
| *No. reflections* | | |
| Used for refinement | 64846 | 23139 |
| Used for Rfree calculation | 3248 | 2310 |
| Completeness (%) | 98.73% | 97.76% |
| Rfactor (%) | 0.1870 (0.4313) | 0.2814 (0.4486) |
| Rfree (%) | 0.2073 (0.4014) | 0.3364 (0.5371) |
| Rwork/Rfree | 0.2079/0.2195 | 0.2814/0.3364 |
| *No. atoms* | | |
| Protein | 1065 | 4078 |
| Water | 124 | 10 |
| *Mean B-factor* | | |
| Protein (Å$^2$) | 27.6 | 87.2 |
| Water (Å$^2$) | 38.5 | 73.4 |
| | | |
| **R.M.S. deviations** | | |
| Bond lengths (Å) | 0.005 | 0.005 |
| Bond angles (º) | 0.839 | 0.84 |
| | | |
| **Molprobity Statistics:** | | |
| Clashscore | 4.18 | 5.85 |
| C-beta deviation | 0 | 0 |
| *Ramachadran Plot* | | |
| Outliers | 0.00% | 0.00% |
| Favored | 100.00% | 97.58% |
| Rotamer Outliers | 0.85% | 3.47% |
| Molprobity Score: | 1.20 | 1.82 |

Statistics related to protein X-ray crystal structures solved in this article. Values in parentheses are for the highest-resolution shell.

# Reporting Summary

## Statistics

For all statistical analyses, confirm that the following items are present in the figure legend, table legend, main text, or Methods section.

| n/a | Confirmed | |
|---|---|---|
| ☐ | ☒ | The exact sample size (*n*) for each experimental group/condition, given as a discrete number and unit of measurement |
| ☐ | ☒ | A statement on whether measurements were taken from distinct samples or whether the same sample was measured repeatedly |
| ☒ | ☐ | The statistical test(s) used AND whether they are one- or two-sided *Only common tests should be described solely by name; describe more complex techniques in the Methods section.* |
| ☐ | ☒ | A description of all covariates tested |
| ☒ | ☐ | A description of any assumptions or corrections, such as tests of normality and adjustment for multiple comparisons |
| ☐ | ☒ | A full description of the statistical parameters including central tendency (e.g. means) or other basic estimates (e.g. regression coefficient) AND variation (e.g. standard deviation) or associated estimates of uncertainty (e.g. confidence intervals) |
| ☒ | ☐ | For null hypothesis testing, the test statistic (e.g. *F*, *t*, *r*) with confidence intervals, effect sizes, degrees of freedom and *P* value noted *Give P values as exact values whenever suitable.* |
| ☒ | ☐ | For Bayesian analysis, information on the choice of priors and Markov chain Monte Carlo settings |
| ☒ | ☐ | For hierarchical and complex designs, identification of the appropriate level for tests and full reporting of outcomes |
| ☐ | ☒ | Estimates of effect sizes (e.g. Cohen's *d*, Pearson's *r*), indicating how they were calculated |

*Our web collection on statistics for biologists contains articles on many of the points above.*

## Software and code

Policy information about availability of computer code

Data collection | Our machine learning models are built in PyTorch 1.11.0 (https://pytorch.org). We make structural predictions of designed sequences with AlphaFold v2.3.1 using localcolabfold v1.5.1 (https://github.com/YoshitakaMo/localcolabfold), ESMFold 2.0.0 (https://github.com/facebookresearch/esm) and OmegaFold v1.1.0 (https://github.com/HeliXonProtein/OmegaFold). Our natural language conditioner makes use of the 125 million parameter GPT-Neo model as available on Hugging Face (https://huggingface.co/EleutherAI/gpt-neo-125m). We construct training datasets based on the PDB (https://www.rcsb.org/), as queried on 2022/03/20, UniProt 2022_01 (https://www.uniprot.org) and PFAM 35 (http://pfam.xfam.org/) . We perform preprocessing with USEARCH (11.0.667) (https://drive5.com/usearch/), mmseq2 13.45111 and pyRosetta 2022.49 (https://www.pyrosetta.org). Our examples of shape conditioning use the Liberation Sans font (https://github.com/liberationfonts/liberation-fonts).

| Data analysis | For data analysis, we use Python 3.9.7 (https://www.python.org), NumPy 1.24.3 (https://numpy.org), Pandas 2.0.2 (https://pandas.pydata.org), matplotlib 3.7.1 (https://matplotlib.org) and seaborn 0.12.2 (https://seaborn.pydata.org). We visualize structures with PyMOL 2.5.0 (https://pymol.org/2). For experimental designs, our nanopore sequencing uses Bonito Basecaller 0.6.1 (https://github.com/nanoporetech/bonito), SeqKit v2.3.1 (https://bioinf.shenwei.me/seqkit), Minimap2 v2.23 (https://github.com/lh3/minimap2), samtools v1.16.1 (https://github.com/samtools/samtools) and pysam v0.20.0 (https://github.com/pysam-developers/pysam). We also use the public BeStSel server (https://bestsel.elte.hu) to analyze circular dichroism data. We calculate TM-scores using the 2019/08/22 version of TM-align (https://zhanggroup.org/TM-align/). Our CATH coverage analysis is based on the CATH S40 4.3 and PDB100 clusters on 2023/08/04 (https://cdn.rcsb.org/resources/sequence/clusters/clusters-by-entity-100.txt) , using Foldseek 5-53465f0 (https://github.com/steineggerlab/foldseek). Our novelty analysis using Gauss integral representations employs the Phaistos suite 1.0 (https://sourceforge.net/projects/phaistos/) and umap-learn 0.5.3 (https://github.com/lmcinnes/umap). We use Stride (https://webclu.bio.wzw.tum.de/stride/) for secondary structure contents. |
|---|---|

For manuscripts utilizing custom algorithms or software that are central to the research but not yet described in published literature, software must be made available to editors and reviewers. We strongly encourage code deposition in a community repository (e.g. GitHub). See the Nature Portfolio guidelines for submitting code & software for further information.

# Data

Policy information about availability of data

All manuscripts must include a data availability statement. This statement should provide the following information, where applicable:
- Accession codes, unique identifiers, or web links for publicly available datasets
- A description of any restrictions on data availability
- For clinical datasets or third party data, please ensure that the statement adheres to our policy

We will not place any restrictions on sharing data from this study. All experimental data, including protein structures that will be deposited in the PDB, will be made available upon publication. All computational results are provided in figures or tables in the main text or supplement.

# Research involving human participants, their data, or biological material

Policy information about studies with human participants or human data. See also policy information about sex, gender (identity/presentation), and sexual orientation and race, ethnicity and racism.

| Reporting on sex and gender | N/A |
|---|---|
| Reporting on race, ethnicity, or other socially relevant groupings | N/A |
| Population characteristics | N/A |
| Recruitment | N/A |
| Ethics oversight | N/A |

Note that full information on the approval of the study protocol must also be provided in the manuscript.

# Field-specific reporting

Please select the one below that is the best fit for your research. If you are not sure, read the appropriate sections before making your selection.

☒ Life sciences    ☐ Behavioural & social sciences    ☐ Ecological, evolutionary & environmental sciences

For a reference copy of the document with all sections, see nature.com/documents/nr-reporting-summary-flat.pdf

# Life sciences study design

All studies must disclose on these points even when the disclosure is negative.

| Sample size | In characterizing Chroma-generated proteins computationally, sample sizes were chosen to be sufficient for estimating distributional properties (e.g., 10,000 or 50,000 generated proteins), such as distributions of secondary structure, contact order, and contact densities. The number of designed proteins for experimental characterization was chosen based on a purposefully pessimistic assumption that well-behaved proteins would occur at a frequency of 1% or higher in unfiltered Chroma distribution. |
|---|---|
| Data exclusions | No data were excluded from analysis in this study. |
| Replication | Split-GFP screens (FACS and Nanopore sequencing) were performed in biological triplicate for unconditional proteins UNC_001 through UNC_172, and in duplicate for unconditional proteins UNC_173 through UNC_268 and proteins conditioned on secondary structure content. |
| Randomization | We did not use randomization because it was not applicable to the study design or analysis. |

| Blinding | We did not use blinding because it was not applicable to the study design or analysis. |

# Reporting for specific materials, systems and methods

We require information from authors about some types of materials, experimental systems and methods used in many studies. Here, indicate whether each material, system or method listed is relevant to your study. If you are not sure if a list item applies to your research, read the appropriate section before selecting a response.

## Materials & experimental systems

| n/a | Involved in the study |
|---|---|
| ☐ ☒ | Antibodies |
| ☒ ☐ | Eukaryotic cell lines |
| ☒ ☐ | Palaeontology and archaeology |
| ☒ ☐ | Animals and other organisms |
| ☒ ☐ | Clinical data |
| ☒ ☐ | Dual use research of concern |
| ☒ ☐ | Plants |

## Methods

| n/a | Involved in the study |
|---|---|
| ☒ ☐ | ChIP-seq |
| ☒ ☐ | Flow cytometry |
| ☒ ☐ | MRI-based neuroimaging |

## Antibodies

| Antibodies used | anti-Strep-tag-HRP (StrepMAB-Classic HRP conjugate, IBA-Lifesciences 2-1509-001) |
|---|---|
| Validation | We observed bands by western blot at the anticipated protein molecular weights using the anti-Strep-tag-HRP antibody with no other background bands observed, and corroborated its specificity using an orthogonal reagent, Streptactin-HRP (IBA-Lifesciences 2-1502-001). Results using these two reagents are compared in Supplementary Fig. 39. |

