## [Peer Review File · Nature]

Manuscript Title: Illuminating protein space with a programmable generative model

Reviewer Comments & Author Rebuttals

Reviewer Reports on the Initial Version:

Referees' comments:

Referee #1:

In the paper. "Illuminating protein space with a programmable generative model", Ingraham et al. present a novel method to generate "meaningful" protein models. The method contains a set of computational advantages over earlier methods (see, for instance, <https://arxiv.org/abs/2206.04119> or <https://arxiv.org/abs/2209.15611>), and follows the progress shown by diffusion models in many areas. The "meaningfulness" (a word used 79 times in the paper) is demonstrated by "in silico" experiments, i.e. designing sequences from the protein model and using OmegaFold to fold these sequences. If at least one of 100 designed sequences folds within 0.5Å TM-score the authors assume that the design is successful.

Unfortunately, the paper does not convince me that this paper provides significant progress over earlier work. In particular, the paper: (1) is lacking any experimental validation that any of the sequences fold to the desired fold; (2) many of the "meaningful" structural models generated do not even pass the (quite loose) in silico test; and (3) from a practical perspective it is unclear that the computational advantages presented here are that useful; in particular, the scaling might not be of great importance as proteins only span about two orders of magnitude in length.

I will detail some comments below:

Major advantages of the paper

Novelty in scalability: A combination of a local and an inverse cubic is novel and could provide an advantage over other methods. However, scaling is probably unimportant as proteins only cover a few orders of magnitude. Methods like AlphaFold can already handle large complexes using bfloat16 and other JAX tricks. The structural model is novel (at least for proteins), but if I understand it correctly, it has about 10x more parameters than the structural model of AlphaFold. Not really clear to me why this is needed or if this can cause problems.

Major

Lack of experimental validation: Many models are not even validated in silico. To the best of my knowledge, no one has experimentally demonstrated that OmegaFold folding strongly correlates with a functional protein. AlphaFold (for a single sequence) has been shown to do this, i.e. it is likely that OmegaFold also could work, but this needs to be verified. At the bare minimum, the authors should use AlphaFold single to verify their in silico tests.

Another problem is that the authors only test a subset of their "meaningful" models in silico. I think this is strongly misleading, and I do think the paper should only mention the models validated in silico. I assume this would remove a large part of the models and (unfortunately) make the paper less appealing.

Some designed models are implausible to fold as predicted, e.g. the shape complementarity. This makes it hard to understand which designs are realistic and which ones are not.

The in silico test uses lax criteria for success (TM-score >0.5). It would be possible to require a structure criterion (TM >0.8). Also unclear if the authors discuss TM-align or TM-score.

The next major problem is code availability. No information is provided, and even after requests from the editor, this reviewer has not been given access to a working copy of the program, i.e. this paper should have been rejected without review. Anyhow, even without the code, this paper raises several questions, and if the code is not released with an open-source licence, the usefulness of this paper drops significantly.

The structural module seems to require about 10x the number of parameters compared with the models from AlphaFold. Is this an advantage?

The network described in Fig. S5 is not well understood. You need to add details about each block.

What are Internal coordinates and Interresidue transforms in Table 2? This lacks detailed descriptions I think.

Hardly any ablation studies are performed.

Minor

The design network is novel, and it would be interesting to compare the performance of this method with ProteinMPNN.

The reference list is not sorted alphabetically, making it very hard to read for this reviewer.

Some details are missing in the description:

How are the initial distributions of distances generated. What is the prior?

Is the graph updated each iteration or not? Does it make a difference?

Is the covariance matrix (big sigma) derived only once from the initial polymer, or is it updated from dewhitened polymer after every iteration?

Can you expand on the advantage of "Variance-Preserving diffusion" (Appendix A)? You claim that it is necessary to obtain a gaussian distribution at time $t=1$, but I did not understand how.

Can you expand on the advantages of diffusion models over (equivariant) normalising flows in sampling protein space?

What are "casually masked GNNs"? Is it the same as in Anand et al., 2022? This needs more details.

It would be interesting to compare your structural module with the one from AlphaFold as ablation studies.

How do you insert the shape-matching loss (section K) to the sampling? Do you compute its gradient of the sum of the two Wasserstein losses and add it to the score?

Referee #2:

In this work, Ingraham et al blend a deep understanding of polymer physics with the practical considerations of common-use cases for protein design to build a computational protein design program, Chroma. The authors use Chroma to generate a set of small and large proteins, and they demonstrate accurate in silico structure prediction for the small proteins. They then apply a series of cleverly crafted external potentials to bias the designed proteins for various design problems, including motif grafting, fold class, symmetry, volumetric shape conditioning, and even a potential based on a natural language model. If Chroma can produce experimentally validated proteins, it would be very useful for the design field. My main concern with the work is that there is no experimental validation of any of the designed proteins. In the absence of any experimentally determined structures, the authors discuss a few nice metrics that demonstrate the “designability” and originality of the generated backbones, such as comparing TERMS with those of natural proteins and set coverage by CATH domains. However, in some cases, e.g., with the shape-conditioned protein alphabet, the authors suggest the models are “plausible” because they have the kinds of secondary structure found in natural proteins. This is too subjective an assessment and could be quantified either in a similar manner as previously mentioned or with predicted structures (Of course, experimental verification would be most welcome.). I am very excited by this work but also have some reservations that I hope the authors can address.

Will the code of Chroma be made freely available? The website demo is insufficient to independently vet the performance of the model, especially since the paper emphasizes conditional generation of structures and this demo offers no such option. How confident are the predicted structures (pLDDT, pAE) of the designed sequences? The authors report T_m score in Fig 3b,c but do not mention model confidence. I used proteinMPNN to compute sequences of some small models (< 140 aa) generated from the Chroma demo and then used AlphaFold2 to fold them in single-sequence mode. The predicted structures agreed with the designed model (approx. 2 Å rmsd) but AF2 was not confident in the structures (pLDDT < 75). Could the authors please discuss a similar kind of assessment of their designed structures?

After reading the manuscript, I still do not fully grasp why the addition of priors from polymer physics is key to success here. I intuitively agree with the statement the authors make in the supplement that the physics-based priors should allow the model to focus on learning the appropriate structure distribution instead of re-learning polymer connectivity. But I do not see that the advantage is quantified in any way other than with these appeals to reason. Is the model faster to train compared to one using uncorrelated gaussian noise? Do the physics-based priors keep the generative model from wandering into unproductive territory more than uncorrelated gaussian noise? The authors describe that a low temperature trick is needed to generate plausible protein structures, which suggests that this approach is not without its nuances. Some additional discussion with quantifiable advantages of the model over other approaches would be welcome in the main text, if possible.

There is not much discussion on the performance of the sequence design model vs the backbone generation model. How important is the new sequence design model to design success? Are there any key differences with published sequence design models such as

pMPNN? Could they use pMPNN sequences for their generated backbones to get better agreement between predicted and generated structures?

As the authors mention, it is exciting that the model can generate proteins within arbitrary fold classes so that designers can begin to focus on designing function. One historical and enabling example is that of the parameterization of coiled coils, which could then be computationally generated in a precisely defined, predictable way. It would be nice to see an example of this kind of precise, fine-tuned generation of backbones for a different class of protein (one not so easily parameterized as coiled coils) enabled by Chroma. This would be an exciting computational result but, despite mentioning this in the text, the authors have stopped short of showing it explicitly.

One main advantage that the authors emphasize in this work is the ability to generate large structures and complexes with low computational cost. It is exciting that Chroma can generate large structures in linear time, but it remains to be seen whether these proteins express and fold to the correct oligomerization state. An experimentally determined structure of a designed protein would be helpful to indicate that the model is able to generate valid structures beyond the breakdown point of structure prediction models such as OmegaFold and AlphaFold2. (How do the predicted structures/sequences compare when folded in AlphaFold2, single-sequence mode?) In the absence of an atomic resolution structure, some evidence that the proteins express and elute at the correct molecular weight on a size-exclusion column would be welcome. For the symmetric complexes, a similar experimental indication that the proteins form the desired oligomerization state would be appropriate.

The authors describe in detail their approach to training a conditional classifier, but a general discussion seems lacking concerning how (or if) to tune the weight of this new gradient relative to the diffusion-learned gradient.

Fig 3b shows results of structure prediction of the designed models. The results are promising for proteins < 300 aa but drop significantly for longer proteins. Each point in the plot is also the best result of 100 designs sequences for that generated backbone. What is the distribution of T_m values over these 100 sequences? Is the distribution in Fig 3c just over the 100 pts shown in Fig 3b, or over the 100 x 100 sequences that were folded with OmegaFold?

The data on which the model is trained should ideally be mentioned in the main text instead of buried in the supplement. Could the authors elaborate on why they train on a small subset of the PDB filtered by homology and sequence similarity? Would it be somehow problematic to train on as much data as possible, while also ensuring that the test, validation, and train sets have minimal overlap? The authors seem to implicitly support this sentiment by adding antibody structures only filtered at 90% sequence similarity (presumably because the CDR loops are the most variable but only account for a fraction of the sequence).

Section E.2 in the supplement could use a detailed figure explaining the computation and operations involved for updating the structure via convex optimization, since this is the key factor that is driving model performance.

Figure 4c takes up a lot of real estate for pictures of protein complexes that have no computational or experimental validation. The authors do not show that any components of these assemblies can be individually predicted and do not design sequences for them; they suggest the structures look plausible because they have helices and sheets. The authors might consider moving this to the supplement to make space for a more substantive description of the model and its essential features.

The authors repeatedly argue that using correlated diffusion is superior to uncorrelated diffusion for the task of backbone generation. Could they quantify this please? One way would be to compare performance of two different models trained with correlated vs uncorrelated, but there are likely other ways to do this without needing to retrain models.

The random graph neural network is an elegant approach to capturing long-distance dependencies without too much additional computational cost relative to kNN alone. In Section D.3, the authors discuss computational complexity and why the random graph neural network has better time/memory performance than more expensive methods such as transformers, but it would be good to see a discussion around model output quality. Does the random graph network perform as well as a fully connected graph network? Better than a simpler kNN-based graph? How much worse? How much better? It would be great to see some quantification of model performance other than the theoretical speed/memory of the computation, if possible. If a model is fast but mostly produces low quality backbones, it might not be as useful as a slower model that produces higher quality structures.

The sequence design network is not fully described in this work. Could the authors please add more detail?

Fig S5, a schematic showing how each of these networks feed into one another would be helpful.

G.2. How long did training take?

There are some typos in the manuscript, e.g., in E.3.

The approach described in K.2 is perhaps best illustrated with a figure that depicts: 1) the goal, 2) the parameters, 3) the approach, and 4) An example of resulting data that is generated and used for the external potential.

Regarding Fig 4b, can the authors please quantify how well the motif-conditioned structures harbor the desired motif? What is the distribution in RMSD for the generated structure vs the input motif? What is this distribution after performing sequence design on the generated

backbone and subsequently predicting its structure? The authors go into much detail in the supplement describing how they encoded the external motif-grafting potential, but they do not fully describe the performance of the model under this potential, outside showing a few panels in Fig 4b.

“The generated structures are once again plausible despite similar difficulties to the DHFR example.” Could the authors please quantify the meaning of “plausible” here?

Did the authors generate the entire capsid complex in Fig. 4a (right) or part of it (from which they then build the full structure from the minimal symmetrized parts)?

Referee #3:

The paper describes a new generative neural network-based model for jointly generating protein sequences and structures. From a purely methodological perspective this is a very interesting paper containing very many novel and extremely interesting ideas, so purely from a methodological viewpoint this is a solid paper.

The main idea is based on the recent success of diffusion models in images; however, unlike existing papers trying to approach protein structure using diffusion models where the noising process generally takes the noised structure very far away from the realm of physically plausible structure, the authors propose a novel diffusion process which means the diffusion process will end up at a random polymer rather than a random cloud of atoms.

The authors further employ graph neural networks with random connectivity to reduce the computational complexity. The main application the authors aim at is protein design.

My main concern, which in my opinion means this is not appropriate for Nature in the current form, is the lack of experimental evaluation.

All evaluation is done for protein design *in silico*, as such it is unclear whether this is applicable to real protein design. If the authors want to make the main point of the paper to be around protein design, there needs to be experimental validation of the designed proteins. This doesn't necessarily need to be full structures however should be indicative of the structures the authors designed the protein for being the structure the protein takes *in vitro*.

The authors use existing protein structure prediction methods (in this case OmegaFold) to compare the predictions of the model to the structure their design model predicts. This has been successfully employed by other groups using AlphaFold as such it's indicative of their method working well; however, this is no replacement for experimental verification.

If the authors do not have the ability to experimentally verify their methods, an alternative way of evaluating their method that would convince me is to evaluate their method as a protein structure prediction method. So the authors should be able to use their method to generate structures conditioned on the amino acid sequences, and if the method works well those should be similar to experimental structures. Seeing if this procedure is competitive to state-of-the-art methods for protein structure prediction e.g. on Cameo would be conclusive evidence for the effectiveness of their method in my opinion. However this would require a reframing of the paper, making it less protein design-centric.

Further I see issues with the authors' claim of subquadratic complexity. From a technical viewpoint this is correct; however I think purely considering runtime is misleading for data-based models. For machine learning models choices such as the random connectivity the authors propose to get subquadratic complexity will in general have an impact on accuracy, so what the authors would need to show beyond just algorithmical complexity is that accuracy does not degrade with length to really make the claim that this is a computationally much more efficient method. In fact, in Fig. 3b the authors show that their generated structures differ much more from OmegaFold predictions as proteins get longer. As the authors point out this could be due to prediction becoming harder; however, it is also possible that their method fails to work at these lengths. The authors would need to demonstrate that this method produces correct results for large systems, i.e. that the particular inductive bias they employ via the sparse connectivity doesn't reduce performance for large systems.

The authors also do not show which of the many novel ideas in this paper contribute to the overall accuracy. The authors should ideally show considerably extended ablations of the different components they introduce.

In a spirit similar to what is currently happening in Image diffusion models the authors show that their model can be conditioned in various ways, including distance-based constraints, forcing the model to generate symmetric structures, conditioning on natural language annotations, etc. This shows that diffusion models provide a flexible framework for protein generation.

Overall I think from a methodological point of view this paper is an excellent contribution to the field; however, I think it is severely held back by the evaluation. I would encourage the authors to extend the evaluation, and with careful evaluation and careful analysis of the contributions I think this can become a landmark paper in the field.

Author Rebuttals to Initial Comments:

We appreciate the reviews' thoughtful feedback and feel that it has helped us to significantly strengthen our work. We highlight four major updates to our manuscript, followed by a more detailed point-by-point response to reviewers:

- 1. Experimental validation of generated proteins.** To demonstrate the designed proteins can express, behave, and fold as intended, we provide a comprehensive set of new experiments validating generated proteins including two high-resolution crystal structures with an average agreement to designs of $\sim 1\text{\AA}$ (**Figure 6, Appendix T, and Supp Figures 31-38**). We screened ~ 310 designs from our model that were not filtered by external methods or *in silico* structure prediction, found many of these proteins express well through a split-GFP assay, and further biophysically characterized a subset of the designs through circular dichromism and DSF assays that suggest that they form stable folds with secondary structure content that is consistent with the designed backbones.
- 2. Systematic ablation study.** To evaluate the impact of novel model components, we trained seven production-scale variants of the model with various ablations and evaluated the resulting impacts on both held-out likelihood (ELBO) and sample quality as measured by large-scale *in silico* refolding (**Appendix K, Supp Fig 18**). We find that our proposed globular covariance models for proteins can improve likelihood over residue gas approaches, that our atomic output layer based on inter-residue geometry prediction can be competitive with and in some cases improve on local-frame update based approaches, and that random long-range graph connections can improve sample quality over a pure k -NN. Additionally, we provide quantitative evidence that our low-temperature sampling method indeed increases sample likelihood and that this is quantitatively associated with sample quality as measured by backbone hydrogen bond formation (**Supp Fig 2**).
- 3. In silico refolding evaluation of all conditional design methods.** To test the plausibility and quality of conditionally generated samples from the model, we performed large scale sampling and refolding tests across our major conditioning methods including substructure, symmetry, shape, class, and natural language (**Appendix J, Supp Figs 12-17**). We find evidence of refolding in every case and across subcases such as varying symmetry groups, varying levels of substructure conditioning context, and varying semantic cues. These results were substantiated by multiple structure prediction methods including AlphaFold, ESMFold, and OmegaFold.

Code and model availability. We now include code and a production environment for sampling backbones and sequences with Chroma, along with a tutorial notebook highlighting each major conditional sampling method. Note that we provide these elements to the reviewers solely for the purpose of carrying out their review. As stated in the licenses included within the code, the reviewers may not use the provided code and weights for anything other than the review process and should destroy any copies they may make after completing their reviews. Note that upon publication of our paper, we will make Chroma available to the research community via a

difference license.

In addition to these broad updates, we would like to provide more detailed responses to each reviewer's comments along with how we have updated the manuscript to address their concerns.

Reviewer 1:

In particular, the paper: (1) is lacking any experimental validation that any of the sequences fold to the desired fold; (2) many of the “meaningful” structural models generated do not even pass the (quite loose) in silico test... The in silico test uses lax criteria for success (TM-score >0.5). It would be possible to require a structure criterion (TM >0.8). ... unclear if the authors discuss TM-align or TM-score.

[See Major Updates 1 & 3]

Thank you for this important point; we agree that there was significantly more validation to do both experimentally and computationally to demonstrate the validity of the proposed structures. We now provide experimental evidence that the designed proteins can fold as intended through two X-ray crystal structures, several CD spectra, and expression characterization for hundreds of designs (**Figure 6**). For our conditional methods that did not previously have *in silico* evidence of refolding, we also provide comprehensive refolding experiments, and we report raw TM scores of refolding as distributions across different conditional methods and subtasks across AlphaFold, ESMFold, and OmegaFold so that no cutoff needs to be applied (**Appendix J**). Lastly, we also clarify in the description of the relevant section that we use TM-align.

(3) from a practical perspective it is unclear that the computational advantages presented here are that useful; in particular, the scaling might not be of great importance as proteins only span about two orders of magnitude in length... A combination of a local and an inverse cubic is novel and could provide an advantage over other methods. However, scaling is probably unimportant as proteins only cover a few orders of magnitude.... It would be interesting to compare your structural module with the one from AlphaFold as ablation studies.

It is an important point that it is *possible* to run AlphaFold for typically sized protein design problems, as well as other cubically- and quadratically-scaling models, and that this is often done in practice. At the same time, sub-quadratic scaling for protein design is still highly desirable; it allows systems to be treated with as large of native context as possible and it can also massively improve the speed, and therefore scale, with which we move through design space. As one recent example, it could be argued that ProteinMPNN saw immediate and widespread adoption because it provides fast sampling via approximately linearly scaling computation with system size.

As a second piece of evidence towards the benefits of linear scaling, we did consider swapping in the AlphaFold structure module for our GNN-based layer for our ablation studies but, because we train on complexes with up to 4000 residues, this was not feasible on our hardware capabilities without significantly reducing the size and complexity of our task or having to concoct some analog of “cropping” that has been used in structure prediction. We consider it a

benefit of our architecture that it is able to train on systems of such a size without having to cut them into smaller subsystems, because this will allow the model to maximally leverage context at design time.

While we did not directly introduce the AlphaFold structure module in our ablations, we did perform ablations measuring the impact of our several new modeling components, which include new diffusion processes, new model architectures, and new sampling algorithms. We find evidence that all quantitatively improve performance of the model (**Appendix K** for ablations and **Appendix B** for low temperature sampling). [See Major Update 2]

I do think the paper should only mention the models validated in silico

With the updated refolding studies (**Appendix J**), we hope to provide evidence that the *methods* are validated, but we still think it is important to examine the raw proposal distributions coming from the model. Otherwise, it can be unclear what performance is coming from the structure prediction method versus what performance is being provided by the underlying generative model. This is also why we intentionally did not use structure prediction refolding as a filter in designing proteins for experimental validation.

The structural module seems to require about 10x the number of parameters compared with the models from AlphaFold. Is this an advantage?

We are unsure which specific part of the model the reviewer is referring to, but if it is the overall parameter count, the main model that we used throughout our experiments is ChromaBackboneA which has ~20 million parameters (**Appendix F**). This is a relatively modest parameter count by contemporary standards; when we downloaded a current version of AlphaFold (via OpenFold) we counted ~93 million parameters.

The network described in Fig. S5 is not well understood. You need to add details about each block.... What are Internal coordinates and Interresidue transforms in Table 2? This lacks detailed descriptions I think... What are “casually masked GNNs”? Is it the same as in Anand et al., 2022? This needs more details.

Thank you for pointing this out; we have since added further descriptions of the model architecture, including an algorithm box for the GNN and a more detailed explanation of the graph features and sequence design network components in **Appendix F**.

The next major problem is code availability.

[See Major Update 4]

We intend to make this model broadly available. Further, we are sharing with the reviewers a preliminary version of the code and a tutorial notebook walking through the sampling methods in this revision package. Importantly, the elements being shared as part of this resubmission are for review purposes, as is stated in the accompanying license file.

The design network is novel, and it would be interesting to compare the performance of this method with ProteinMPNN.

We have added an evaluation in **Supp Fig 11** and shown that our method's performance is comparable to ProteinMPNN as measured by sequence recovery.

How are the initial distributions of distances generated. What is the prior? Is the covariance matrix (big sigma) derived only once from the initial polymer, or is it updated from dewhitened polymer after every iteration? ... Can you expand on the advantage of "Variance-Preserving diffusion" (Appendix A)? You claim that it is necessary to obtain a gaussian distribution at time $t=1$, but I did not understand how.

To answer the reviewer's specific questions, the prior is a Gaussian distribution over Cartesian coordinates with zero mean and a covariance matrix from our set of proposed covariance matrices. The base covariance matrix is stationary and only a function of sequence length. The distances we discuss in **Appendix C** are the distances induced by this distribution, and we note that an interesting property of the Variance Preserving diffusion is that it will preserve quadratic forms, which includes the preservation of squared distance statistics. So, if there are certain squared distances which have the same expected value in the data distribution and in a given prior, these squared distances will be preserved in expectation through the corresponding diffusion process.

We have found visualizations helpful to get a better sense of the connection between the prior and the corresponding diffusion processes, and so we have added new figures to the manuscript including visualizations comparing the proposed diffusion priors and processes in **Supp Fig. 3** as well as a visualization of the underlying prior covariance matrix itself along with conditional (clamped) samples in **Supp Fig 19**.

Can you expand on the advantages of diffusion models over (equivariant) normalising flows in sampling protein space?

Technically, when treated with the Probability Flow ODE formulation, our model is an equivariant normalizing flow over protein space. This follows from Song et al.'s result that continuous time diffusion models can be treating as continuous normalizing flows without retraining (**Appendix A.5**) and that our architecture is equivariant (**Appendices E and F**).

That said, the training of conventional normalizing flows (whether continuous or discrete-time) has been difficult because it requires backpropagating through entire simulation trajectories with $O(T)$ memory and compute cost, where T is the number of time steps. A major benefit of the denoising diffusion paradigm is that we can directly sample from, and also train on, instantaneous samples from any time point along the diffusion process, thus reducing those costs to be $O(1)$ in both memory and time. As a result of this "simulation-free" training, we can train much higher capacity models for the same GPU budget.

Is the graph updated each iteration or not? Does it make a difference?...

We dynamically update the graph at every time step of the diffusion process during both training time and testing time. Interestingly, another useful side effect of the simulation-free $O(1)$ training cost (previous comment) of denoising diffusion models is that we can sample random graphs at training time without having to worry about backpropagating through graph construction. Without this dynamic updating, the graph topologies from high-noise states would be applied to the low noise states and likely not include residue contacts that form during the reverse diffusion.

How do you insert the shape-matching loss (section K) to the sampling? Do you compute its gradient of the sum of the two Wasserstein losses and add it to the score?

We directly add the gradient of the ShapeLoss with a time-dependent scaling and have added a section describing this scaling in **Appendix Q**.

Reviewer 2:

Will the code of Chroma be made freely available?

[See Major Update 4]

Yes, we intend to make this model broadly available. Further, we are sharing with the reviewers a preliminary version of the code and a tutorial notebook walking through the sampling methods in this revision package. Importantly, the elements being shared as part of this resubmission are for review purposes, as is stated in the accompanying license file.

How confident are the predicted structures (pLDDT, pAE) of the designed sequences? The authors report Tm score in Fig 3b,c but do not mention model confidence. I used proteinMPNN to compute sequences of some small models (< 140 aa) generated from the Chroma demo and then used AlphaFold2 to fold them in single-sequence mode. The predicted structures agreed with the designed model (approx. 2 Å rmsd) but AF2 was not confident in the structures (pLDDT < 75). Could the authors please discuss a similar kind of assessment of their designed structures?

We were very glad to hear that the model could work in the reviewer's hands. Regarding the general point about refolding and model confidence, while we were adding our new refolding experiments and ablation studies [Major Updates 2 & 3], we also have investigated how TM score and confidence related across multiple checkpoints and folding algorithms (**Supp Fig 17**). We find that they are strongly, though not perfectly, correlated and agree that confidence would likely be a useful and potentially differentiated filter in practice for design.

After reading the manuscript, I still do not fully grasp why the addition of priors from polymer physics is key to success here. I intuitively agree with the statement the authors make in the supplement that the physics-based priors should allow the model to focus on learning the appropriate structure distribution instead of re-learning polymer connectivity. But I do not see that the advantage is quantified in any way other than with these appeals to reason. Is the model faster to train compared to one using uncorrelated gaussian noise? Do the physics-based priors keep the generative model from wandering into unproductive territory more than uncorrelated gaussian noise? The authors describe

that a low temperature trick is needed to generate plausible protein structures, which suggests that this approach is not without its nuances. Some additional discussion with quantifiable advantages of the model over other approaches would be welcome in the main text, if possible.... The authors repeatedly argue that using correlated diffusion is superior to uncorrelated diffusion for the task of backbone generation. Could they quantify this please? One way would be to compare performance of two different models trained with correlated vs uncorrelated, but there are likely other ways to do this without needing to retrain models.

Thank you for this comment; we now have evidence to support our correlated diffusion based on polymer covariance. In our new ablation experiments (**Appendix K**), we trained seven models representing different configurations of proposed methodology, two of which were based on a Gaussian “Residue Gas”, chain-uncorrelated Gaussian covariance model. Instead of being fully uncorrelated, which would cause atoms even within a residue to become dissociated from each other, we design a block diagonal covariance matrix that captures spatial proximity between atoms *within* a residue. We consider this to be a close Gaussian surrogate to C-alpha diffusions and “Frame diffusion” presented in *Trippe et al* and *Anand et al*, but with the added benefit that it can also model all backbone atomic degrees of freedom and capture non-ideality. We explain this covariance model in **Appendix C** and present a visualization comparing the different diffusions with **Supp Fig 3**.

In our ablation experiments (**Appendix K**), we learn several things, including:

- Almost all model configurations can produce backbone samples that are designable and refold *in silico*.
- From the point of view of likelihood, Globular Covariance is favorable to Residue Gas covariance
- From the point of sample quality as measured by refolding, random graphs are favorable to k-NN based graphs

Ultimately, while all proposed model components showed potential improvements to either likelihood and/or sample quality in some capacity, the single aspect of our framework on which sample quality most critically rests is our novel low temperature sampling algorithm. We have added further quantitative evidence showing that this method does successfully sample from high likelihood states in **Supp Fig 2** and that these states are associated with improved sample quality as measure by hydrogen bonding rates (i.e., secondary structure content). We note that this low-temperature sampling algorithm is entirely general and in fact has been applied by multiple other groups to achieve strong results while our work was on a preprint server [1,2].

[1] <https://arxiv.org/abs/2304.03889>

[2] <https://arxiv.org/abs/2304.05364>

There is not much discussion on the performance of the sequence design model vs the backbone generation model. How important is the new sequence design model to design success? Are there any key differences with published sequence design models such as pMPNN? Could they use pMPNN sequences for their generated backbones to get better agreement between predicted and generated structures?... The sequence design network is not fully described in this work. Could the authors please add more detail?

We have added an evaluation comparing different variants of our ChromaDesign model and ProteinMPNN and showing they have comparable performance as measured by sequence recovery in **Supp Fig 11**. We also have added further details on the architecture of the model in **Appendix F** and on the low-complexity penalties we use for sampling in **Appendix H**.

As the authors mention, it is exciting that the model can generate proteins within arbitrary fold classes so that designers can begin to focus on designing function. One historical and enabling example is that of the parameterization of coiled coils, which could then be computationally generated in a precisely defined, predictable way. It would be nice to see an example of this kind of precise, fine-tuned generation of backbones for a different class of protein (one not so easily parameterized as coiled coils) enabled by Chroma. This would be an exciting computational result but, despite mentioning this in the text, the authors have stopped short of showing it explicitly.

We think this is an excellent idea and are quite excited about the prospect of being able to robustly index a particular family of folds. While we agree that finding a particular small number of parameters that quantitatively “organize” a family remains an open question, we do see repeated generation of diverse samples subject to constraints in our substructural infilling method **Supp Fig 20**, for which we see a high degree of refolding in **Supp Fig 12**. The routine satisfaction of diverse samples subject to constraints with realistic protein geometries suggests that we are very close to parametrically-indexable families (one could consider our “plane-cut” families a family in some sense).

The authors describe in detail their approach to training a conditional classifier, but a general discussion seems lacking concerning how (or if) to tune the weight of this new gradient relative to the diffusion-learned gradient.

Thank you for raising this point - balancing the level of classifier guidance has certainly been a common problem in the diffusion modeling literature and our work is no exception. We have added some explanations of how we tune the guidance scale and gradient clipping cutoffs for neural network classifiers in **Appendix J.4** and also include the guidance scale as a hyperparameter in our refolding studies of semantically conditioned samples in **Supp Fig 16**.

Fig 3b shows results of structure prediction of the designed models. The results are promising for proteins < 300 aa but drop significantly for longer proteins. Each point in the plot is also the best result of 100 designs sequences for that generated backbone. What is the distribution of Tm values over these 100 sequences? Is the distribution in Fig 3c just over the 100 pts shown in Fig 3b, or over the 100 x 100 sequences that were folded with Omegafold?

To clarify, **Fig. 3c** is indeed the distribution of the same points as shown in **Fig. 3b**. To give a broader sense of the raw TM scores across different tasks and folding algorithms we now include a systematic refolding study in **Appendix J and Supp Figs 12-17** and in the ablation study **Supp Fig 18** which show the values TM scores (sometimes best of k , sometimes raw).

The data on which the model is trained should ideally be mentioned in the main text instead of buried in the supplement. Could the authors elaborate on why they train on a small subset of the PDB filtered by homology and sequence similarity? Would it be somehow problematic to train on as much data as possible, while also ensuring that the test, validation, and train sets have minimal overlap? The authors seem to implicitly support this sentiment by adding antibody structures only filtered at 90% sequence similarity (presumably because the CDR loops are the most variable but only account for a fraction of the sequence).

This is an important point that we would like to make clearer in the text. In our experience we agree we would ideally train on *all* the data in the PDB, and the main issue has been that generative models are extremely sensitive to the distributional weighting of different classes in the training set. Training on all of the data in the PDB with a flat weighting will make the model very sensitive to “overrepresented folds” in the PDB, e.g. trypsins, kinases, certain model viral proteins, etc. Ideally, we would train on all of these data as a “stream” from which structures are drawn relative to their uniqueness to up-weight underrepresented structures. Our current approximation to that is simply redundancy reduction. We note this problem is far more of an issue for generative models than for deterministic structure prediction models because they are estimating a marginal distribution instead of estimating a conditional mode.

Section E.2 in the supplement could use a detailed figure explaining the computation and operations involved for updating the structure via convex optimization, since this is the key factor that is driving model performance.

We agree and have added a figure to better explain and visualize the inter residue-based geometry prediction in **Supp Fig 5**.

Figure 4c takes up a lot of real estate for pictures of protein complexes that have no computational or experimental validation. The authors do not show that any components of these assemblies can be individually predicted and do not design sequences for them; they suggest the structures look plausible because they have helices and sheets. The authors might consider moving this to the supplement to make space for a more substantive description of the model and its essential features.

If this comment is specifically in relation to the shape-conditioning section, the main point that we hoped to not bury is that diffusion models can solve for geometries subject to surprising and complex constraints. We also have some evidence that shape conditioning for complex geometries can produce structures that refold *in silico* (**Supp Fig 14**). But the point is taken and we will consider reorganizing more of the novel evaluation results into the main figures.

The random graph neural network is an elegant approach to capturing long-distance dependencies without too much additional computational cost relative to kNN alone. In Section D.3, the authors discuss computational complexity and why the random graph neural network has better time/memory performance than more expensive methods such as transformers, but it would be good to see a discussion around model output quality.

Does the random graph network perform as well as a fully connected graph network? Better than a simpler kNN-based graph? How much worse? How much better? It would be great to see some quantification of model performance other than the theoretical speed/memory of the computation, if possible. If a model is fast but mostly produces low quality backbones, it might not be as useful as a slower model that produces higher quality structures.

We thank the reviewer for the kind words and agree that the random graph neural network idea warrants further investigation. While training on fully connected graphs is not feasible with our current hardware setup without significantly reducing the maximum complex size (our current largest complex is 4000 residues), we do compare random graphs vs k-NN graphs in our ablation study (**Appendix K, Supp Fig 18**) and find that the random long range graph connections improve performance over a pure k-NN based approach as measured by AlphaFold refolding.

Regarding Fig 4b, can the authors please quantify how well the motif-conditioned structures harbor the desired motif? What is the distribution in RMSD for the generated structure vs the input motif? What is this distribution after performing sequence design on the generated backbone and subsequently predicting its structure? The authors go into much detail in the supplement describing how they encoded the external motif-grafting potential, but they do not fully describe the performance of the model under this potential, outside showing a few panels in Fig 4b.

Since our original submission, we have discovered a better method for substructure conditioning and focused most of our evaluation on it. The new method is based on hard constraints rather than soft restraints by implementing a modified Langevin dynamics, i.e. a formulation where it satisfies the substructure conditioning by construction. This new method, described in **Appendix M**, combines low temperature annealed Langevin dynamics with a custom covariance (inverse mass) matrix that assigns infinite mass to the “clamped” regions of structure. The challenge in this formulation is to allow the model to rapidly explore in the constrained conformational space (e.g., unclamped structure / loop space), i.e. to take the benefit of coherent chain motions while also satisfying boundary constraints. Fortunately, we can do this precisely with our globular covariance prior by applying Gaussian conditioning formulas, as visualized in **Supp Fig 19**. We find that this infilling method achieves high refolding rates across a variety of conditioning scenarios as shown in **Supp Fig 12**.

“The generated structures are once again plausible despite similar difficulties to the DHFR example.” Could the authors please quantify the meaning of “plausible” here?

The word plausible here was imprecise, and we now focus on infilling sample quality through refolding as shown in **Supp Fig 12**.

Did the authors generate the entire capsid complex in Fig. 4a (right) or part of it (from which they then build the full structure from the minimal symmetrized parts)?

To generate symmetric assemblies, we introduce a novel approach (**Algorithm 4**) in which, at each step of the SDE solver, we compute the drift and diffusion terms for a randomly selected

contiguous subset of the entire structure. To achieve this, we employ a process of randomly sampling an asymmetric unit and identifying substructures based on the distance-based k -nearest neighbors (k -NN). Throughout the entire trajectory of the SDE, all asymmetric units contribute to the update of the SDE solver through a process called symmetric broadcasting. Therefore, this technique allows the algorithm to effectively have global awareness while maintaining a constant memory cost with respect to symmetry orders. The capsid-like structures depicted in **Figure 4** were generated using a value of $k = 6$, so the model never needs to put the entire 60 subunits (960,000 atoms) into memory. Our algorithm can generate refoldable protein assemblies validated with AlphaFold Multimer (**Appendix J.2** and **Supp Fig 13**), demonstrating the effectiveness of our approach.

Reviewer 3:

The paper describes a new generative neural network-based model for jointly generating protein sequences and structures. From a purely methodological perspective this is a very interesting paper containing very many novel and extremely interesting ideas, so purely from a methodological viewpoint this is a solid paper.

We thank the reviewer for the kind words and are very glad to hear that some of the ideas may be of broader interest.

My main concern, which in my opinion means this is not appropriate for Nature in the current form, is the lack of experimental evaluation. ... All evaluation is done for protein design in silico, as such it is unclear whether this is applicable to real protein design. If the authors want to make the main point of the paper to be around protein design, there needs to be experimental validation of the designed proteins. This doesn't necessarily need to be full structures however should be indicative of the structures the authors designed the protein for being the structure the protein takes in vitro. The authors use existing protein structure prediction methods (in this case OmegaFold) to compare the predictions of the model to the structure their design model predicts. This has been successfully employed by other groups using AlphaFold as such it's indicative of their method working well; however, this is no replacement for experimental verification.

[See Major Updates 1 & 3]

We agree that confirmation of real proteins behaving as intended in the lab is an important proof point, and we have updated the manuscript to now include two X-ray crystal structures that agree with models at $\sim 1\text{\AA}$, thermal melt and circular dichroism data to characterize several proteins that melt and show secondary structure content consistent with design, and expression data for 310 designs that suggest a large fraction can express well in a bacterial system (**Figure 6**, **Appendix T**, and **Supp Figures 31-38**).

Of course, there are still many more potential applications for the model than can be tested in the lab in the context of a single study. Thus, we sought to characterize our major conditioning methods by systematic assessment of refolding under multiple methods and for multiple subtasks (**Appendix J** and **Supp Figs 12-17**). As described in our major update, we find evidence of refolding in all cases and for structures at a variety of lengths and secondary structure content.

The authors also do not show which of the many novel ideas in this paper contribute to the overall accuracy. The authors should ideally show considerably extended ablations of the different components they introduce.

[See Major Updates 2]

We have added an ablation study evaluating the impact of these different ideas on model performance as measured by held out likelihood and sample quality in (**Appendix K, Supp Fig 18**). We modify the model along several dimensions including: (1) choice of covariance model, (2) choice of graph type, (3) choice of atomic output parameterization layer, and (3) choice of loss function(s). We find that all presented components can contribute to favorable performance in different ways, however, the most essential aspect of the framework for being able to generate high-quality samples appears to be our low temperature sampling algorithm. We have further evidence that low-temperature sampling drives the model towards high likelihood states (**Supp Fig 2**), and we also find that these high likelihood states are strongly associated with increased secondary structure formation.

Further I see issues with the authors' claim of subquadratic complexity. From a technical viewpoint this is correct; however I think purely considering runtime is misleading for data-based models. For machine learning models choices such as the random connectivity the authors propose to get subquadratic complexity will in general have an impact on accuracy, so what the authors would need to show beyond just algorithmical complexity is that accuracy does not degrade with length to really make the claim that this is a computationally much more efficient method. In fact, in Fig. 3b the authors show that their generated structures differ much more from OmegaFold predictions as proteins get longer. As the authors point out this could be due to prediction becoming harder; however, it is also possible that their method fails to work at these lengths. The authors would need to demonstrate that this method produces correct results for large systems, i.e. that the particular inductive bias they employ via the sparse connectivity doesn't reduce performance for large systems.

The distinction between the complexity required for an algorithm and the complexity required for effective performance is an important one and we appreciate that the reviewer has raised this. First, to calibrate expectations, we want to make clear that we would generally expect a fully connected system to perform as well as or better than a sparsely connected system if it is well-tuned and the data are at least of modest size. In that sense we think of our Random GNN architectures as an analog of Sparse Transformers (discussed in **Appendix D**).

Unfortunately, with our current hardware capabilities, we are unable to train quadratically scaling models in any reasonable amount of time without significantly changing the task and approach, as our task includes protein complexes of up to 4000 residues in size. Indeed our ChromaBackboneA model, which has only 60 edges per node, took 10 weeks to train on 8 V100 GPUs. While we were not able to measure the cost of approximating a fully connected system with a sparse graph, we were able to at least compare the approaches of a purely local sparse graph vs a graph that allocated some of the edges towards random long-range connections. We

found that, as measured by refolding under AlphaFold, the random graph approach was favorable to a purely local k-NN (*Appendix K* and *Supp Fig 18*).

So, it remains to be seen what the cost of approximating fully connected architectures with sub-quadratic, graph-based approaches will be for design larger protein systems. We do think that this work provides some of the first evidence (to our knowledge) that a sub-quadratic graph-based approach can continue to perform well even for some large systems, e.g. the symmetry refolding experiments in *Supp Fig 13*.

In a spirit similar to what is currently happening in Image diffusion models the authors show that their model can be conditioned in various ways, including distance-based constraints, forcing the model to generate symmetric structures, conditioning on natural language annotations, etc. This shows that diffusion models provide a flexible framework for protein generation. ... Overall I think from a methodological point of view this paper is an excellent contribution to the field; however, I think it is severely held back by the evaluation. I would encourage the authors to extend the evaluation, and with careful evaluation and careful analysis of the contributions I think this can become a landmark paper in the field.

We thank this reviewer for their thoughtful and encouraging remarks, and we too are very excited by the prospect of diffusion models in the field of protein design. We hope that our added evaluations, both experimental and computational, will relieve the reviewers' main reservations regarding the previous version of our manuscript.

Reviewer Reports on the First Revision:

Referees' comments:

Referee #1:

With the experimental validations and clarifications, this paper has strongly improved and could in principle be accepted (although I think a shorter version of the paper might be more suitable). Obviously, it is very discouraging that the authors claim to "we will make Chroma available to the research community via a difference license" (I assume they mean different). It can be noted that the authors have had several months to decide what license to use, so the plan means actually nothing. However, this is in the end an editorial decision if this purposely vague statement is sufficient for acceptance. (If I was the editor I would demand an open-source license and enforce that the paper was out before publication.)

Referee #2:

The authors provide an updated manuscript with experimental validation of their diffusion model. They have addressed all my questions and concerns, and I believe the work is high quality.

I am glad to read that the authors will release the code. I look forward to seeing it out. I was not able to access the code itself for review. But I was able to run the model in the notebook, and I am satisfied. From the revision, it is clear that evaluating backbone likelihood (ELBO) should be considered when generating backbones, so I hope there will be an obvious way to perform this calculation within the code, given a set of input coordinates.

It is interesting that there were no strongly correlated variables with model expression/solubility. The authors might want to check for (if they haven't already) the presence of any N-terminal degrons within their designed sequences (e.g., Leu and Phe for E. coli).

The authors provide much welcomed ablation studies to evaluate important elements of their model performance. While it does not seem from the loss function that random graphs are better than kNN graphs, it is clear from the folding calculations that random graphs produce more predictably "foldable" backbones, so this approach will likely see wide adoption in the community.

The authors allow diffusion of atoms within a residue to account for non-ideality, although they did not analyze its importance (e.g., in the two crystal structures, were there any residues that deviated from ideality that agreed with the diffused backbone coordinates?). This non-ideality seems potentially important, and it sets this model apart from current popular models such as RFDiffusion.

Fig. S34: There is a protein in "mostly beta" and a protein in "mixed alpha+beta" that appears to be an all-alpha protein.

Referee #3:

The authors have substantially improved the paper by including experimental validation and also providing ablations of the various components of their algorithm. The authors also include results showing that the model is comparable in terms of accuracy to other contemporary methods (ProteinMPNN), while being in a very different part of the neural network architecture space, which makes this very interesting.

Overall the authors' experimental validation seems thorough including X-ray crystal structures for

two of their designs, showing that they reproduce the structures the authors designed for to high accuracy.

There are a few comments that I think would be useful to address however:

In Fig. 4 the authors show their experiments on symmetry-conditioned, substructure-conditioned and shape-conditioned sampling; however I am concerned that for the shape-conditioned samples in Appendix J3 the authors show very low success rates when refolding with structure prediction methods and no experimental validation, so I'm highly doubtful if those are meaningful. I think they are interesting and should be mentioned, but I don't think they should be displayed so prominently.

The same applies to the higher order symmetries. Here the authors only showed refolding results for trimer subsets, so there are questions regarding how well those reflect reality. Again these are interesting results and should be included in the paper; however given these caveats they shouldn't be displayed so prominently in my opinion.

I would suggest leaving both of these to the appendix.

Author Rebuttals to First Revision:

Referee 1:

it is very discouraging that the authors claim to "we will make Chroma available to the research community via a difference license" (I assume they mean different). It can be noted that the authors have had several months to decide what license to use, so the plan means actually nothing.

We apologize for our delay in aligning on the final software license, and we agree on the importance of using a standardized open-source license so that the community may build on the work in an unrestricted manner. As we mention above, we are happy to now commit to the Apache 2.0 license for the code, along with a non-commercial license on the model weights.

Referee 2:

was able to run the model in the notebook, and I am satisfied. From the revision, it is clear that evaluating backbone likelihood (ELBO) should be considered when generating backbones, so I hope there will be an obvious way to perform this calculation within the code, given a set of input coordinates.

We are glad to hear that you were able to use the model in our notebook environment. We will be releasing the code, along with a notebook environment like this and tools for computing metrics such as the ELBO in an open-source repository on GitHub with an Apache 2.0 license. The weights will be accessible under a non-commercial license.

It is interesting that there were no strongly correlated variables with model expression/solubility. The authors might want to check for (if they haven't already) the presence of any N-terminal degrons within their designed sequences (e.g., Leu and Phe for E. coli).

We agree that the lack of a clear signal here is interesting. While the concern about degrons would seem important for our *E. coli* system, our expression system involves a standardized N-terminal polyhistidine tag so it would be less clear how to test for any more subtle affects beyond the first few residues.

The authors allow diffusion of atoms within a residue to account for non-ideality, although they did not analyze its importance (e.g., in the two crystal structures, were there any residues that deviated from ideality that agreed with the diffused backbone coordinates?). This non-ideality seems potentially important, and it sets this model apart from current popular models such as RFdiffusion.

Thank you for making this note about the model's ability to capture non-ideal states, which we also feel is an important aspect of the framework. While we were considering doing analyses like this, it became clear that the vast majority of bond length and angle information in the PDB (and potentially in our own models) is likely driven primarily by model building software due to limited data resolution. As a result,

making this point convincingly would seem to require careful curation of many high-resolution structures along with more involved analysis of the underlying electron density maps. We leave this to future work.

Referee 3:

In Fig. 4 the authors show their experiments on symmetry-conditioned, substructure-conditioned and shape-conditioned sampling; however I am concerned that for the shape-conditioned samples in Appendix J3 the authors show very low success rates when refolding with structure prediction methods and no experimental validation, so I'm highly doubtful if those are meaningful. I think they are interesting and should be mentioned, but I don't think they should be displayed so prominently.

The same applies to the higher order symmetries. Here the authors only showed refolding results for trimer subsets, so there are questions regarding how well those reflect reality. Again these are interesting results and should be included in the paper; however given these caveats they shouldn't be displayed so prominently in my opinion.

I would suggest leaving both of these to the appendix.

We agree that, based on the refolding analyses in our June revision, the Shape-conditioned and potentially some of the higher-order symmetry conditioned sampling protocols would seem to have mixed to low evidence for their plausibility of generating productive designs. Fortunately, as part of our model standardization efforts (major point 1 above), we recomputed all of our refolding evaluations with the final version of our model, which was the one that showed the most promise in the ablation study. We observe that this final model version shows significantly improved rates of refolding across both shape and symmetry conditioning tasks (**Appendix J Supp. Figure 17-18**), where we now observe widespread refolding of all letters and numbers across three structure prediction methods and can furnish a discernable alphabet purely of models predicted by ESMfold (**Supp. Figure 18**). We have regenerated all main text figures with samples from these finalized protocols, and also reduced the sizes of the largest symmetry examples in the main figure.

We wish to thank the editor and reviewers for all their helpful and encouraging feedback on this work; it has significantly shaped the course of the paper in a way that we hope makes it more broadly useful to the community.

Sincerely,

The Authors

Gevorg Grigoryan, PhD
Chief Technology Officer
Generate Biomedicines, Inc.

Generate Biomedicines

101 South Street, Suite 900, Somerville, MA 02143

www.generatebiomedicines.com

Generate:Biomedicines
A Flagship Pioneering Company

Research Associate Professor
Computer Science and Biology
Dartmouth College